# Accelerating Non-Conjugate Gaussian Processes By Trading Off Computation For Uncertainty

**Lukas Tatzel**                                                    *lukas.tatzel@uni-tuebingen.de*
*University of Tübingen, Tübingen AI Center*

**Jonathan Wenger**                                                    *jw4246@columbia.edu*
*Columbia University*

**Frank Schneider**                                                    *f.schneider@uni-tuebingen.de*
*University of Tübingen, Tübingen AI Center*

**Philipp Hennig**                                                    *philipp.hennig@uni-tuebingen.de*
*University of Tübingen, Tübingen AI Center*

**Reviewed on OpenReview:** *https://openreview.net/forum?id=UdcF3JbSKb*

## Abstract

Non-conjugate Gaussian processes (NCGPs) define a flexible probabilistic framework to model categorical, ordinal and continuous data, and are widely used in practice. However, exact inference in NCGPs is prohibitively expensive for large datasets, thus requiring approximations in practice. The approximation error adversely impacts the reliability of the model and is not accounted for in the uncertainty of the prediction. We introduce a family of iterative methods that explicitly model this error. They are uniquely suited to parallel modern computing hardware, efficiently recycle computations, and compress information to reduce both the time and memory requirements for NCGPs. As we demonstrate on large-scale classification problems, our method significantly accelerates posterior inference compared to competitive baselines by trading off reduced computation for increased uncertainty.

## 1 Introduction

Non-conjugate Gaussian processes[1] (NCGPs) form a fundamental interpretable model class widely used throughout the natural and social sciences. For example, NCGPs are applied to count data in biomedicine, categorical data in object classification tasks, and continuous data in time series regression. An NCGP assumes the data is generated from an exponential family likelihood with a Gaussian process (GP) prior over the latent function. Such a *probabilistic* approach is essential in domains where critical decisions must be made based on limited information, such as in public policy, medicine or robotics.

Unfortunately, even the conjugate Gaussian case, where fitting an NCGPs reduces to GP regression, naively has cubic time complexity $\mathcal{O}(N^3)$ in the number of training data $N$ and requires $\mathcal{O}(N^2)$ memory, which is prohibitive for modern large-scale datasets. For non-Gaussian likelihoods, inference has to be done approximately, which generally exacerbates this problem. For example, inference via the Laplace approximation (LA) boils down to finding the mode of the log posterior via Newton's method, which is equivalent to solving a *sequence* of regression problems (Bishop, 2006; MacKay, 1992; Spiegelhalter & Lauritzen, 1990).

---

[1]Such a model is also called Generalized Gaussian Process Model (Chan & Dong, 2011) or Generalized Linear Model (Nelder & Wedderburn, 1972). The latter name is sometimes used only for latent Gaussian models, which can lead to confusion. The models studied in this work are generally of *nonparametric* nature. The resulting large linear problems are the main reason why the algorithms we propose are relevant in the first place.

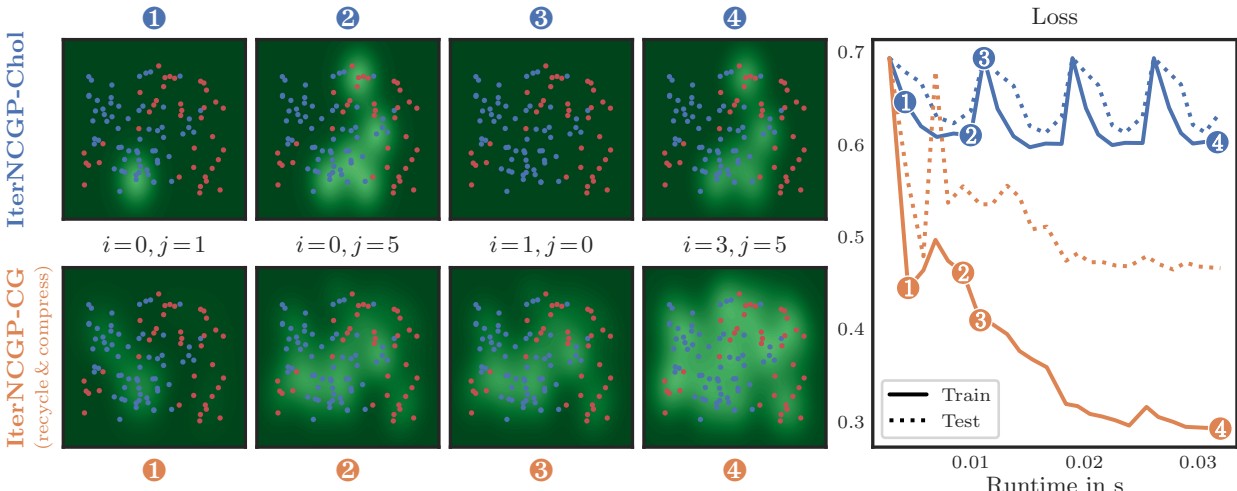

**Figure 1: Binary Classification with IterNCGP.** Comparison of two IterNCGP variants: *(Top)* IterNCGP variant corresponding to data subsampling and solving each regression problem exactly in each Newton step $i$. *(Bottom)* IterNCGP variant with a more informative policy (details in Section 3.2), recycling of computations between Newton steps (details in Section 3.3) and compression to reduce memory (details in Section 3.4). The panels show the marginal uncertainty ( ) over the latent function at Newton step $i$ and solver iteration $j$. Using recycling, the current belief is efficiently propagated between mode-finding steps $i$ (❷→❸) without performance drops *(Right)*. Details in Appendix C.1.

Due to limited computational resources, large-scale problems often require approximations. The resulting error affects a model's predictive accuracy but also its uncertainty quantification. Hence, the question arises: **Can NCGPs be efficiently trained on extensive data without compromising reliability?**

Recently, iterative methods have emerged which in the conjugate Gaussian case allow an explicit, tunable trade-off between reduced computation and increased uncertainty (Trippe et al., 2019; Wenger et al., 2022b). This computational uncertainty quantifies the inevitable approximation error in the sense of probabilistic numerics (Cockayne et al., 2019b; Hennig et al., 2015; 2022; Oates & Sullivan, 2019).

**Contributions.** In this work, we take a similar approach and extend Wenger et al. (2022b)'s IterGP (that assumes a *conjugate* Gaussian likelihood) to *non-conjugate* exponential family likelihoods. This is a non-trivial extension, as the posterior is no longer Gaussian and *multiple* related regression problems have to be solved. Specifically, we propose **(i)** IterNCGP: a family of efficient inference algorithms for NCGPs with a tunable trade-off between computational savings and added uncertainty (Section 3.1) with **(ii)** mechanisms to tailor the inference algorithm to a specific downstream application (Section 3.2). In response to the specific computational challenges in the non-conjugate setting, we develop **(iii)** novel strategies to optimally recycle costly computations (Section 3.3) and **(iv)** to restrict the memory usage, with minimal impact on inference (Section 3.4).

Our algorithm IterNCGP consists of two nested loops: An outer loop (indexed by $i$) which iterates over Newton steps/GP regression problems, each of which is solved approximately via an inner loop (indexed by $j$) that implements a probabilistic linear solver. Figure 1 shows the marginal uncertainty over the latent function at different stages of that process and illustrates the effectiveness of (ii), (iii) and (iv). Specifically, by recycling computations between Newton steps, we are able to traverse the two-loop structure "diagonally" which leads to steady progress without performance drops between Newton steps.

## 2  Background

Let $(\boldsymbol{X}, \boldsymbol{y})$ be a dataset of $N$ input vectors $\{\boldsymbol{x}_n \in \mathbb{X}\}_{n=1}^N$ stacked into $\boldsymbol{X} = (\boldsymbol{x}_1, \ldots \boldsymbol{x}_N)^\top \in \mathbb{R}^{N \times D}$ and corresponding outputs $\boldsymbol{y} = (y_1, \ldots, y_N)^\top \in \mathbb{Y}^N$, where $\mathbb{X} = \mathbb{R}^D$ and $\mathbb{Y} = \mathbb{R}$ or $\mathbb{Y} = \mathbb{N}_0$ (regression) or $\mathbb{Y} = \{1, \ldots, C\}$ (classification).

## 2.1 Non-conjugate Gaussian Processes (NCGPs)

We consider the probabilistic model $p(\boldsymbol{y}, \boldsymbol{f} \mid \boldsymbol{X}) = p(\boldsymbol{y} \mid \boldsymbol{f})\, p(\boldsymbol{f} \mid \boldsymbol{X})$, where the vector $\boldsymbol{f} := f(\boldsymbol{X}) \in \mathbb{R}^{NC}$ is given by a latent function $f \colon \mathbb{X} \to \mathbb{R}^{C}$ evaluated at the training data.

**Prior.** Assume a multi-output Gaussian process prior $\mathcal{GP}(m, K)$ over the latent function with mean function $m \colon \mathbb{X} \to \mathbb{R}^{C}$ and kernel function $K \colon \mathbb{X} \times \mathbb{X} \to \mathbb{R}^{C \times C}$. Therefore the latent vector has density $p(\boldsymbol{f} \mid \boldsymbol{X}) = \mathcal{N}(\boldsymbol{f}; \boldsymbol{m}, \boldsymbol{K})$ with mean $\boldsymbol{m} := m(\boldsymbol{X}) \in \mathbb{R}^{NC}$ and covariance $\boldsymbol{K} = K(\boldsymbol{X}, \boldsymbol{X}) \in \mathbb{R}^{NC \times NC}$ defined by $N^2$ blocks $K(\boldsymbol{x}_i, \boldsymbol{x}_j) \in \mathbb{R}^{C \times C}$. Each such block represents the covariance between the $C$ latent functions evaluated at inputs $\boldsymbol{x}_i$ and $\boldsymbol{x}_j$.

**Likelihood.** Assume iid data, which depends on the latent function via an inverse link function $\lambda \colon \mathbb{R}^{C} \to \mathbb{R}^{C}$, s.t. $p(\boldsymbol{y} \mid \boldsymbol{f}) = \prod_{n=1}^{N} p(y_n \mid \lambda(\boldsymbol{f}_n))$, where $p(y_n \mid \lambda(\boldsymbol{f}_n))$ is a log-concave likelihood, e.g. any exponential family distribution.[2] For example, for Poisson regression the inverse link function is given by $\lambda(\boldsymbol{f}_n) = \exp(\boldsymbol{f}_n)$ and for multi-class classification by $\lambda(\boldsymbol{f}_n) = \mathrm{softmax}(\boldsymbol{f}_n)$.

For nonlinear inverse link functions, the posterior $p(f \mid \boldsymbol{X}, \boldsymbol{y})$ and predictive distribution $p(y_\diamond \mid \boldsymbol{X}, \boldsymbol{y}, \boldsymbol{x}_\diamond) = \int p(y_\diamond \mid f_\diamond) p(f_\diamond \mid \boldsymbol{X}, \boldsymbol{y}, \boldsymbol{x}_\diamond)\, df_\diamond$ are computationally intractable, requiring approximations.

## 2.2 Approximate Inference via Laplace

A popular way to perform approximate inference in an NCGP is to use a Laplace approximation (LA) (Bishop, 2006; MacKay, 1992; Spiegelhalter & Lauritzen, 1990). The idea is to approximate the posterior

$$p(\boldsymbol{f} \mid \boldsymbol{X}, \boldsymbol{y}) \approx q(\boldsymbol{f} \mid \boldsymbol{X}, \boldsymbol{y}) := \mathcal{N}(\boldsymbol{f}; \boldsymbol{f}_{\mathrm{MAP}}, \boldsymbol{\Sigma}), \tag{1}$$

with a Gaussian with mean given by the mode $\boldsymbol{f}_{\mathrm{MAP}}$ of the log-posterior and covariance $\boldsymbol{\Sigma} := -(\nabla^2 \log p(\boldsymbol{f}_{\mathrm{MAP}} \mid \boldsymbol{X}, \boldsymbol{y}))^{-1}$ given by the negative inverse Hessian (with respect to $\boldsymbol{f}$) at the mode. Due to the assumed GP prior over the latent function, the log-posterior is given by

$$\Psi(\boldsymbol{f}) := \log p(\boldsymbol{f} \mid \boldsymbol{X}, \boldsymbol{y}) \overset{\mathrm{c}}{=} \log p(\boldsymbol{y} \mid \boldsymbol{f}) + \log p(\boldsymbol{f} \mid \boldsymbol{X}) \overset{\mathrm{c}}{=} \log p(\boldsymbol{y} \mid \boldsymbol{f}) - \frac{1}{2}(\boldsymbol{f} - \boldsymbol{m})^\top \boldsymbol{K}^{-1}(\boldsymbol{f} - \boldsymbol{m}) \tag{2}$$

We use $\overset{\mathrm{c}}{=}$ to denote equality up to an additive constant.

**Mode-Finding via Newton's Method.** To find the mode $\boldsymbol{f}_{\mathrm{MAP}}$, one typically uses Newton steps, i.e.

$$\boldsymbol{f}_{\mathrm{MAP}} \approx \boldsymbol{f}_{i+1} = \boldsymbol{f}_i - \nabla^2 \Psi(\boldsymbol{f}_i)^{-1} \cdot \nabla \Psi(\boldsymbol{f}_i), \tag{3}$$

where $\nabla \Psi(\boldsymbol{f}_i) = \nabla \log p(\boldsymbol{y} \mid \boldsymbol{f}_i) - \boldsymbol{K}^{-1}(\boldsymbol{f}_i - \boldsymbol{m})$ and $\nabla^2 \Psi(\boldsymbol{f}_i) = -\boldsymbol{W}(\boldsymbol{f}_i) - \boldsymbol{K}^{-1}$. The negative Hessian $\boldsymbol{W}(\boldsymbol{f}_i) := -\nabla^2 \log p(\boldsymbol{y} \mid \boldsymbol{f}_i)$ of the log likelihood at $\boldsymbol{f}_i$ is positive definite for all $\boldsymbol{f}$, since we assumed a log-concave likelihood. Therefore $\Psi$ is concave and the Newton updates are well-defined.

## 2.3 Predictions

Using a local quadratic Taylor approximation of the log-posterior $\Psi$ around the current iterate $\boldsymbol{f}_i$, we obtain the LA $\mathcal{N}(\boldsymbol{f}; \boldsymbol{f}_{i+1}, -\nabla^2 \Psi(\boldsymbol{f}_i)^{-1})$ whose mean is given by the maximizer of the local quadratic, i.e. the subsequent Newton iterate $\boldsymbol{f}_{i+1}$. Substituting this in place of the posterior, the predictive distribution for the latent function $p(f(\cdot) \mid \boldsymbol{X}, \boldsymbol{y}) = \int p(f(\cdot) \mid \boldsymbol{f}) \mathcal{N}(\boldsymbol{f}; \boldsymbol{f}_{i+1}, -\nabla^2 \Psi(\boldsymbol{f}_i)^{-1})\, d\boldsymbol{f}$ is a Gaussian process $\mathcal{GP}(m_{i,*}, K_{i,*})$, with mean and covariance functions

$$m_{i,*}(\cdot) := m(\cdot) + K(\cdot, \boldsymbol{X})\boldsymbol{K}^{-1}(\boldsymbol{f}_{i+1} - \boldsymbol{m}), \tag{4}$$

$$K_{i,*}(\cdot, \cdot) := K(\cdot, \cdot) - K(\cdot, \boldsymbol{X})\hat{\boldsymbol{K}}(\boldsymbol{f}_i)^{-1}K(\boldsymbol{X}, \cdot), \tag{5}$$

where $\hat{\boldsymbol{K}}(\boldsymbol{f}_i) := \boldsymbol{K} + \boldsymbol{W}(\boldsymbol{f}_i)^{-1}$ (cf. Eq. (3.24); Rasmussen & Williams, 2006). We obtain the predictive distribution for $y_\diamond$ at test input $\boldsymbol{x}_\diamond$ by integrating this approximative posterior against the likelihood, i.e.

---

[2]The Hessian of an exponential family likelihood is the negative Hessian of its log-partition function, which equals the *positive definite* covariance matrix of its sufficient statistics.

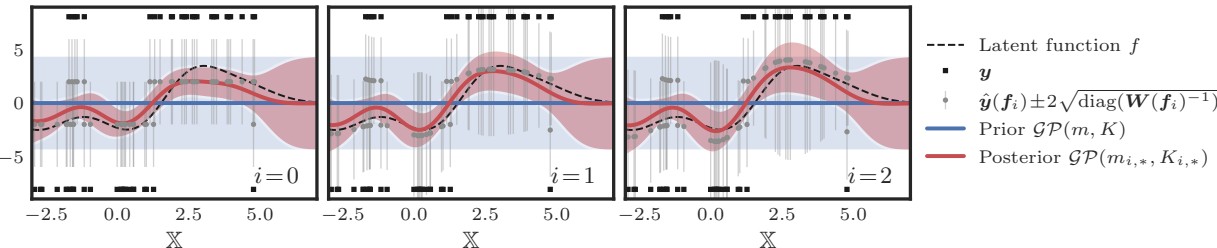

**Figure 2: Approximate Inference in NCGPs as Sequential GP Regression.** Performing a LA at a Newton iterate $\boldsymbol{f}_i$ results in a posterior GP that coincides with the posterior to a GP regression problem with pseudo targets $\hat{\boldsymbol{y}}(\boldsymbol{f}_i)$ observed with Gaussian noise $\mathcal{N}(\boldsymbol{0}, \boldsymbol{W}(\boldsymbol{f}_i)^{-1})$. The plot shows an illustration of this connection for binary classification on a toy problem with the latent function drawn from a GP. Notice how similar the posteriors are between Newton steps. This motivates our proposed strategy for recycling computations between steps in Section 3.3. Details in Appendix C.1.

$p(y_\diamond \mid \boldsymbol{X}, \boldsymbol{y}, \boldsymbol{x}_\diamond) = \int p(y_\diamond \mid \boldsymbol{f}_\diamond)\, p(\boldsymbol{f}_\diamond \mid \boldsymbol{X}, \boldsymbol{y}, \boldsymbol{x}_\diamond)\, d\boldsymbol{f}_\diamond$. This $C$-dimensional integral can be approximated via quadrature, MC-sampling or specialized approaches (like the probit method (MacKay, 1992) for a categorical likelihood and softmax inverse link function).

## 3 Computation-Aware Inference in NCGPs

While Newton's method typically converges in a few steps for a log-concave likelihood, each step in (3) requires linear system solves with symmetric positive (semi-)definite matrices of size $NC \times NC$. Naively computing these solves via Cholesky decomposition is problematic even for moderately sized datasets due to its cubic time $\mathcal{O}(N^3 C^3)$ and quadratic memory complexity $\mathcal{O}(N^2 C^2)$. We will demonstrate in the following how to circumvent this issue by reducing the computations in exchange for increased uncertainty about the latent function.

### 3.1 Derivation of the IterNCGP Framework

**Overview.** As a first step, we reinterpret the posterior predictive mean (Equation (4)) as the GP posterior for a specific regression problem (Equation (6)). The sequence of regression problems is connected to the sequence of Newton steps and forms the outer loop of our algorithm ITERNCGP, indexed by $i$ (Algorithm 1). The formulation as GP regression problem allows us to apply ITERGP (Wenger et al., 2022b) as an inner loop, indexed by $j$ (Algorithm 2). By using ITERGP, we can solve each regression problem approximately and quantify the resulting error in the form of additional uncertainty (Equation (9)).

**Outer Loop: Newton's Method as Sequential GP Regression.** Through the LA, we obtain a posterior predictive $f \sim \mathcal{GP}(m_{i,*}, K_{i,*})$ over the latent function for each Newton step. As we show in Appendix A.1, the posterior predictive (Equations (4) and (5)) in step $i$ can be written as

$$m_{i,*}(\cdot) = m(\cdot) + K(\cdot, \boldsymbol{X})\hat{\boldsymbol{K}}(\boldsymbol{f}_i)^{-1}(\hat{\boldsymbol{y}}(\boldsymbol{f}_i) - \boldsymbol{m}) \tag{6}$$

$$K_{i,*}(\cdot, \cdot) = K(\cdot, \cdot) - K(\cdot, \boldsymbol{X})\hat{\boldsymbol{K}}(\boldsymbol{f}_i)^{-1}K(\boldsymbol{X}, \cdot), \tag{7}$$

where $\hat{\boldsymbol{y}}(\boldsymbol{f}_i) := \boldsymbol{f}_i + \boldsymbol{W}(\boldsymbol{f}_i)^{-1}\nabla \log p(\boldsymbol{y} \mid \boldsymbol{f}_i)$. Equations (6) and (7) have the exact form of the posterior for a GP regression problem with fictitious *pseudo targets* $\hat{\boldsymbol{y}}(\boldsymbol{f}_i)$ observed with Gaussian noise $\mathcal{N}(\boldsymbol{0}, \boldsymbol{W}(\boldsymbol{f}_i)^{-1})$.[3] Figure 2 shows an illustration of this interpretation.

Equation (6) requires solving a linear system $\hat{\boldsymbol{K}}(\boldsymbol{f}_i)\, \boldsymbol{v} = \hat{\boldsymbol{y}}(\boldsymbol{f}_i) - \boldsymbol{m}$ of size $NC \times NC$. The posterior mean is then simply given by $m_{i,*}(\cdot) = m(\cdot) + K(\cdot, \boldsymbol{X})\boldsymbol{v}$. However, also the Newton update from Equation (3)

---

[3]If $\boldsymbol{W}(\boldsymbol{f}_i)^{-1}$ does not exist, e.g. in multi-class classification, we substitute its pseudo-inverse $\boldsymbol{W}(\boldsymbol{f}_i)^{\dagger}$, which for multi-class classification can be evaluated efficiently (see Appendix A.6). Alternatively, one can place a prior on the sum of the $C$ latent functions (MacKay, 1998, Eq. (10)).

---

**Algorithm 1: IterNCGP Outer loop.**

---

**Input:** GP prior $\mathcal{GP}(m, K)$, training data $(\boldsymbol{X}, \boldsymbol{y})$, $\nabla p(\boldsymbol{y}|\boldsymbol{f}, \boldsymbol{X})$ and access to products with $\boldsymbol{K}$ and $\boldsymbol{W}(\boldsymbol{f})^{-1}$
**Output:** GP posterior $\mathcal{GP}(m_{i,j}, K_{i,j})$

| | | | **Time** | **Memory** |
|---|---|---|---|---|
| 1 | **procedure** IterNCGP$(m, K, \boldsymbol{X}, \boldsymbol{y}, \boldsymbol{f}_0 = \boldsymbol{m})$ | | | |
| 2 | $\quad \boldsymbol{m} \leftarrow m(\boldsymbol{X}),$ | $\boldsymbol{m}$: Prior mean vector | $\mathcal{O}(\tau_{\boldsymbol{m}})$ | $\mathcal{O}(NC)$ |
| 3 | $\quad$ Provide access to $\boldsymbol{w} \mapsto \boldsymbol{Kw}$ | $\boldsymbol{K}$: Prior covariance/kernel matrix | | $\mathcal{O}(\mu_{\boldsymbol{K}})$ |
| 4 | $\quad$ Initialize buffers $\boldsymbol{S}, \boldsymbol{T} \in \mathbb{R}^{NC \times 0}$ | $\boldsymbol{S}$: actions, $\boldsymbol{T}$: products with $\boldsymbol{K}$ | | |
| 5 | $\quad$ **for** $i = 0, 1, 2, \ldots$ **while not** OuterStoppingCriterion() **do** | | | |
| 6 | $\quad\quad$ Provide access to $\boldsymbol{w} \mapsto \boldsymbol{W}(\boldsymbol{f}_i)^{-1}\boldsymbol{w}$ | $\boldsymbol{W}(\boldsymbol{f}_i)^{-1}$: Observation noise | | $\mathcal{O}(\mu_{\boldsymbol{W}^{-1}})$ |
| 7 | $\quad\quad \hat{\boldsymbol{y}}(\boldsymbol{f}_i) \leftarrow \boldsymbol{f}_i + \boldsymbol{W}(\boldsymbol{f}_i)^{-1}\nabla \log p(\boldsymbol{y}|\boldsymbol{f}_i)$ | $\hat{\boldsymbol{y}}(\boldsymbol{f}_i)$: Pseudo targets | $\mathcal{O}(\tau_{\boldsymbol{W}^{-1}} + NC)$ | $\mathcal{O}(NC)$ |
| 8 | $\quad\quad \mathcal{GP}(m_{i,j}, K_{i,j}), \boldsymbol{v}_j \leftarrow$ IterGP$(m, K, \boldsymbol{X}, \boldsymbol{y}, \boldsymbol{m}, \boldsymbol{K}, \boldsymbol{W}(\boldsymbol{f}_i)^{-1}, \hat{\boldsymbol{y}}(\boldsymbol{f}_i), \boldsymbol{S}, \boldsymbol{T})$ | | | |
| 9 | $\quad\quad \boldsymbol{f}_{i+1} \leftarrow \boldsymbol{Kv}_j + \boldsymbol{m}$ | Approximate Newton update | $\mathcal{O}(\tau_{\boldsymbol{K}} + NC)$ | $\mathcal{O}(NC)$ |
| 10 | $\quad$ **return** $\mathcal{GP}(m_{i,j}, K_{i,j})$ | | | |

---

The IterGP algorithm is given in Algorithm 2. Instructions in blue are needed for recycling (see Section 3.3). The matrices $\boldsymbol{K}$ and $\boldsymbol{W}^{-1}(\boldsymbol{f}_i)$ are evaluated lazily. We thus report the runtime costs when the matrix-vector products are actually computed. For an in-depth discussion of the computational costs, see Appendix B.3.

---

follows directly from $\boldsymbol{v}$ since $\boldsymbol{f}_{i+1} = \boldsymbol{Kv} + \boldsymbol{m}$. In that sense, computing the posterior predictive mean and performing Newton updates are equivalent. The sequence of Newton steps/GP regression problems forms the outer loop of our algorithm IterNCGP. The pseudo code is given in Algorithm 1.

**Inner Loop: Computation-Aware GP Regression via IterGP.** Reframing the Newton iteration as sequential GP regression does not yet solve the need for linear solves with a matrix of size $NC \times NC$. However, it allows us to leverage recent advances for GP regression, specifically the IterGP algorithm introduced by Wenger et al. (2022b). IterGP is matrix-free, i.e. only relies on matrix-vector products $\boldsymbol{s} \mapsto \hat{\boldsymbol{K}}(\boldsymbol{f}_i)\boldsymbol{s}$, reducing the required memory from quadratic to linear, and efficiently exploits modern parallel GPU hardware (Charlier et al., 2021).

Internally, IterGP uses a probabilistic linear solver (PLS) (Cockayne et al., 2019a; Hennig, 2015; Wenger & Hennig, 2020) to iteratively compute a Gaussian belief over the so-called *representer weights*, i.e. over the solution of the linear system $\hat{\boldsymbol{K}}(\boldsymbol{f}_i)\boldsymbol{v} = \hat{\boldsymbol{y}}(\boldsymbol{f}_i) - \boldsymbol{m}$. In each solver iteration, indexed by $j$, the belief $\mathcal{N}(\boldsymbol{v}; \boldsymbol{v}_j, \boldsymbol{\Omega}_j)$ is updated by conditioning the Gaussian on a one-dimensional projection $\alpha_j = \boldsymbol{s}_j^\top \boldsymbol{r}_{j-1}$ of the preceding residual $\boldsymbol{r}_{j-1} = \hat{\boldsymbol{y}}(\boldsymbol{f}_i) - \boldsymbol{m} - \hat{\boldsymbol{K}}(\boldsymbol{f}_i)\boldsymbol{v}_{j-1}$. The vector $\boldsymbol{s}_j \leftarrow$ Policy() is called an *action* and is generated by a user-specified *policy*. The policy determines the behavior of the solver by "weighting" the residual $\boldsymbol{r}_{j-1}$ for specific data points (we discuss the role of the policy in the context of NCGPs in Section 3.2.). Wenger et al. (2022b, Tab. 1) lists policies that have classic counterparts, e.g. unit vectors $\boldsymbol{s}_j \leftarrow \boldsymbol{e}_j$ correspond to partial Cholesky and residual actions $\boldsymbol{s}_j \leftarrow \boldsymbol{r}_{j-1}$ to conjugate gradients (Hestenes & Stiefel, 1952).

The belief over the representer weights translates into an approximate GP posterior over the latent function,

$$m_{i,j}(\cdot) := m(\cdot) + K(\cdot, \boldsymbol{X})\boldsymbol{v}_j \tag{8}$$

$$K_{i,j}(\cdot, \cdot) := K(\cdot, \cdot) - K(\cdot, \boldsymbol{X})\boldsymbol{C}_j K(\boldsymbol{X}, \cdot), \tag{9}$$

where $\boldsymbol{v}_j = \boldsymbol{C}_j(\hat{\boldsymbol{y}}(\boldsymbol{f}_i) - \boldsymbol{m})$. Crucially, by Wenger et al. (2022b, Thm. 2), the posterior covariance in Equation (9) *exactly* quantifies the error in each approximate Newton step introduced by only using *limited computational resources*, i.e. running the linear solver for $j \ll NC$ iterations. This reduces the time complexity to $\mathcal{O}(jN^2C^2)$. The approximate precision matrix in Equations (8) and (9),

$$\boldsymbol{C}_j = \boldsymbol{S}_j(\boldsymbol{S}_j^\top \hat{\boldsymbol{K}}(\boldsymbol{f}_i)\boldsymbol{S}_j)^{-1}\boldsymbol{S}_j^\top \tag{10}$$

with $\boldsymbol{S}_j = (\boldsymbol{s}_1, \ldots, \boldsymbol{s}_j) \in \mathbb{R}^{NC \times j}$ has rank $j$ and approaches $\hat{\boldsymbol{K}}(\boldsymbol{f}_i)^{-1}$ as $j \to NC$. Intuitively, it projects $\hat{\boldsymbol{K}}(\boldsymbol{f}_i)$ onto the subspace spanned by the actions $\boldsymbol{S}_j$, then inverts and projects the result back into the original space. The IterGP algorithm is given in Algorithm 2.

**The Marginal Uncertainty Decreases in the Inner Loop.** As we perform more solver iterations, the marginal uncertainty captured by the posterior covariance (Equation (9)) contracts, i.e. for each $i$ it holds

---

**Algorithm 2: IterNCGP Inner Loop: IterGP with a Virtual Solver Run.**

---

**Input:** GP prior $\mathcal{GP}(m, K)$, training data $(\boldsymbol{X}, \boldsymbol{y})$, $\boldsymbol{m}$, access to products with $\boldsymbol{K}$ and $\boldsymbol{W}^{-1}$, pseudo targets $\hat{\boldsymbol{y}}$, buffers $\boldsymbol{S}, \boldsymbol{T}$

**Output:** GP posterior $\mathcal{GP}(m_{i,j}, K_{i,j})$

| | | | **Time** | **Memory** |
|---|---|---|---|---|
| 1 | **procedure** IterGP$(m, K, \boldsymbol{X}, \boldsymbol{y}, \boldsymbol{m}, \boldsymbol{K}, \boldsymbol{W}^{-1}, \hat{\boldsymbol{y}}, \boldsymbol{S}, \boldsymbol{T})$ | | | |
| 2 | $\boldsymbol{C}_0, \boldsymbol{S}, \boldsymbol{T} \leftarrow$ VirtualSolverRun$(\boldsymbol{S}, \boldsymbol{T}, \boldsymbol{W}^{-1})$  See Algorithm 3 | | | |
| 3 | $\boldsymbol{v}_0 \leftarrow \boldsymbol{C}_0(\hat{\boldsymbol{y}} - \boldsymbol{m})$    $\boldsymbol{v}_0$ : Consistent initial iterate | | $\mathcal{O}(RNC)$ | $\mathcal{O}(NC)$ |
| 4 | **for** $j = 1, 2, 3, \dots$ **while not** InnerStoppingCriterion() **do** | | | |
| 5 | $\boldsymbol{r}_{j-1} \leftarrow (\hat{\boldsymbol{y}} - \boldsymbol{m}) - \boldsymbol{K}\boldsymbol{v}_{j-1} - \boldsymbol{W}^{-1}\boldsymbol{v}_{j-1}$    $\boldsymbol{r}_{j-1}$ : Residual vector | | $\mathcal{O}(\tau_{\boldsymbol{K}} + \tau_{\boldsymbol{W}^{-1}} + NC)$ | $\mathcal{O}(NC)$ |
| 6 | $\boldsymbol{s}_j \leftarrow$ Policy()    Select action $\boldsymbol{s}_j$ via policy | | $\mathcal{O}(\tau_{\text{POLICY}})$ | $\mathcal{O}(NC)$ |
| 7 | Append $\boldsymbol{s}_j$ to buffer $\boldsymbol{S} \leftarrow (\boldsymbol{S}, \boldsymbol{s}_j) \in \mathbb{R}^{NC \times B}$ | | | $\mathcal{O}(BNC)$ |
| 8 | $\alpha_j \leftarrow \boldsymbol{s}_j^\top \boldsymbol{r}_{j-1}$    $\alpha_j$ : Observation is projection of residual onto action | | $\mathcal{O}(NC)$ | $\mathcal{O}(1)$ |
| 9 | $\boldsymbol{t}_j \leftarrow \boldsymbol{K}\boldsymbol{s}_j$    First term in $\hat{\boldsymbol{K}}\boldsymbol{s}_j = \boldsymbol{K}\boldsymbol{s}_j + \boldsymbol{W}^{-1}\boldsymbol{s}_j$ | | $\mathcal{O}(\tau_{\boldsymbol{K}})$ | $\mathcal{O}(NC)$ |
| 10 | Append $\boldsymbol{t}_j$ to buffer $\boldsymbol{T} \leftarrow (\boldsymbol{T}, \boldsymbol{t}_j) \in \mathbb{R}^{NC \times B}$ | | | $\mathcal{O}(BNC)$ |
| 11 | $\boldsymbol{z}_j \leftarrow \boldsymbol{t}_j + \boldsymbol{W}^{-1}\boldsymbol{s}_j$    Second term in $\hat{\boldsymbol{K}}\boldsymbol{s}_j = \boldsymbol{K}\boldsymbol{s}_j + \boldsymbol{W}^{-1}\boldsymbol{s}_j$ | | $\mathcal{O}(\tau_{\boldsymbol{W}^{-1}} + NC)$ | $\mathcal{O}(NC)$ |
| 12 | $\boldsymbol{d}_j \leftarrow \boldsymbol{s}_j - \boldsymbol{C}_{j-1}\boldsymbol{z}_j$    $\boldsymbol{d}_j$ : Search direction | | $\mathcal{O}(BNC)$ | $\mathcal{O}(NC)$ |
| 13 | $\eta_j \leftarrow \boldsymbol{z}_j^\top \boldsymbol{d}_j$    $\eta_j$ : Normalization constant | | $\mathcal{O}(NC)$ | $\mathcal{O}(1)$ |
| 14 | $\boldsymbol{Q}_j \leftarrow (\boldsymbol{Q}_{j-1}, 1/\sqrt{\eta_j}\,\boldsymbol{d}_j) \in \mathbb{R}^{NC \times B}$    Append column | | $\mathcal{O}(NC)$ | $\mathcal{O}(BNC)$ |
| 15 | $\boldsymbol{C}_j \leftarrow \boldsymbol{Q}_j \boldsymbol{Q}_j^\top$    Rank $B$ approximation $\boldsymbol{C}_j \approx \hat{\boldsymbol{K}}^{-1}$ | | | |
| 16 | $\boldsymbol{v}_j \leftarrow \boldsymbol{v}_{j-1} + \frac{\alpha_j}{\eta_j}\boldsymbol{d}_j$    $\boldsymbol{v}_j$ Updated representer weights estimate | | $\mathcal{O}(NC)$ | $\mathcal{O}(NC)$ |
| 17 | $m_{i,j}(\cdot) \leftarrow m(\cdot) + K(\cdot, \boldsymbol{X})\boldsymbol{v}_j$    Equation (8) | | $\mathcal{O}(NN_\diamond C^2)$ | $\mathcal{O}(N_\diamond C)$ |
| 18 | $K_{i,j}(\cdot, \cdot) \leftarrow K(\cdot, \cdot) - K(\cdot, \boldsymbol{X})\boldsymbol{C}_j K(\boldsymbol{X}, \cdot)$    Equation (9) | | $\mathcal{O}(B(N + N_\diamond)N_\diamond C^2)$ | $\mathcal{O}(N_\diamond^2 C^2)$ |
| 19 | **return** $\mathcal{GP}(m_{i,j}, K_{i,j})$ and $\boldsymbol{v}_j$ | | | |

Instructions in blue are needed for recycling (see Section 3.3). $\boldsymbol{C}_j$ is represented via its root $\boldsymbol{Q}_j$ and evaluated lazily. We thus report the runtime costs when the matrix-vector products are actually computed. The costs for evaluating the posterior GP $\mathcal{GP}(m_{i,j}, K_{i,j})$ are based on $N_\diamond$ test data points $\boldsymbol{X}_\diamond \in \mathbb{R}^{N_\diamond \times D}$. For an in-depth discussion of the computational costs, see Appendix B.3.

---

(element-wise) that $\text{diag}(K_{i,j}(\boldsymbol{x}, \boldsymbol{x})) \geqslant \text{diag}(K_{i,k}(\boldsymbol{x}, \boldsymbol{x}))$ for any $k \geqslant j$ and arbitrary $\boldsymbol{x}$. This is because the approximate precision matrix $\boldsymbol{C}_j$ grows in rank with each solver iteration. For a detailed derivation, see Appendix A.3.

**Summary.** Finding the posterior mode $\boldsymbol{f}_{\text{MAP}}$ and the corresponding predictive distributions (Equations (4) and (5)) can be viewed from different angles. Through an optimization lens, we use Newton updates, each maximizing a local quadratic approximation of the log-posterior. From a probabilistic perspective, we solve a sequence of *related* GP regression problems and IterGP enables us to propagate a probabilistic estimate of the latent function throughout the *entire* optimization process.

For Gaussian likelihoods, the LA (Equation (1)) is exact and a single Newton step suffices. Consequently, our framework generalizes IterGP to arbitrary log-concave likelihoods (Theorem A.2). We now explore the role of the policy and its potential for actions tailored to specific problems (Section 3.2). We also leverage the relatedness of GP regression problems in the outer loop for further speedups (Section 3.3) and introduce a mechanism to control IterNCGP's memory usage (Section 3.4).

### 3.2 Policy Choice: Targeted Computations

Algorithm 2 defines a *family* of inference algorithms. Its instances, defined by a concrete action policy, generally behave quite differently. To better understand what effect the sequence of actions $\boldsymbol{S}_j = (\boldsymbol{s}_1, \dots, \boldsymbol{s}_j) \in \mathbb{R}^{NC \times j}$ has on IterNCGP, we consider the following examples.

**Unit Vector Policy = Subset of Data (SoD).** Choosing the actions $\boldsymbol{s}_j = \boldsymbol{e}_j$ to be unit vectors with all zeros except for a one at entry $j$, corresponds to (sequentially) conditioning on the first $j$ data points in the training data in each GP regression subproblem, since $K(\cdot, \boldsymbol{X})\boldsymbol{S}_j = K(\cdot, \boldsymbol{X}_{1:j})$. Therefore this policy is equivalent to simply using a subset $\boldsymbol{X}_{1:j} \in \mathbb{R}^{j \times D}$ of the data and performing *exact* GP regression (e.g. via

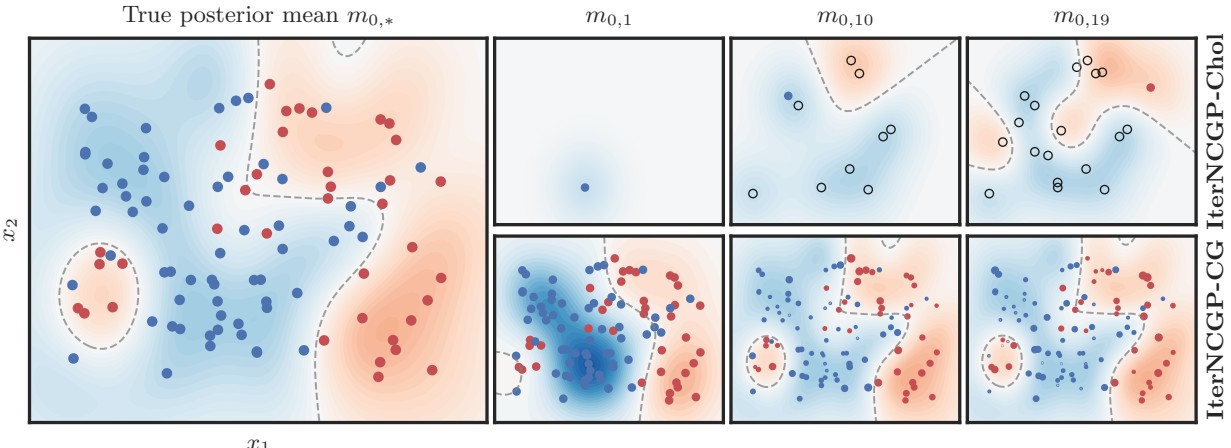

**Figure 3: Different IterNCGP Policies Applied to GP Classification.** *(Left)* The true posterior mean $m_{0,*}$ (▮) for binary classification (●/●) and its decision boundary (- -). *(Right)* Current posterior mean estimate after $1, 10$, and $19$ iterations with the unit vector policy (*Top*) and the CG policy (*Bottom*). Shown are the data points selected by the policy in this iteration with the dot size indicating their relative weight. For IterNCGP-Chol, data points are targeted one by one and previously used data points are marked with (○). Details in Appendix C.1

a Cholesky decomposition) in each Newton iteration in Algorithm 1. This basic policy shows how actions *target computation* as illustrated in the top row of Figure 3.

**(Conjugate) Gradient Policy.** Instead of targeting individual data points, we can also specify *weighted* linear combinations of them to target the data more globally. E.g. , using the current residual $s_j = r_{j-1} = \hat{y}(f_i) - m - \hat{K}(f_i)v_{j-1}$, approximately targets those data most, where the posterior mean prediction is far off.[4] As Wenger et al. (2022b) show, this corresponds to using conjugate gradients (CG) (Hestenes & Stiefel, 1952) to estimate the posterior mean. This policy is illustrated in the bottom row of Figure 3.

### 3.3 Recycling: Reusing Computations

Using IterGP with a suitable policy for GP inference allows us to solve each GP regression problem more efficiently. However, for NCGP inference, we must solve *multiple* regression problems—one per mode-finding step. Figure 2 suggests that GP posteriors across steps are highly similar. Leveraging this observation, we develop a novel approach, designed specifically for the NCGP setting, that *efficiently recycles costly computations* between outer loop steps (pseudo code in Algorithm 3).

The cost of IterNCGP is dominated by repeated matrix-vector products with $K$ (see Section 3.5). However, these costly operations can be recycled and used over multiple Newton steps: Consider the matrix-vector products with an action vector $s$ in the first and second mode-finding step as an example:

$$\text{Step } i = 0: \qquad s \mapsto \hat{K}(f_0)s = Ks + W(f_0)^{-1}s$$
$$\text{Step } i = 1: \qquad s \mapsto \hat{K}(f_1)s = Ks + W(f_1)^{-1}s.$$

Since $K$ is independent of $f_i$, the product $Ks$ is *shared* among both operations.

**Virtual Solver Run.** Assume we have used $B$ action vectors $(s_1, \dots, s_B) =: S \in \mathbb{R}^{NC \times B}$ in step $i = 0$, and buffered the matrix-vector products $(Ks_1, \dots, Ks_B) = KS =: T$. In the next Newton step $i = 1$ we apply the *same* actions to a *new* linear system of equations. From Equation (10) we obtain

$$C = SM^{-1}S^\top \text{ with } M := S^\top(KS + W(f_1)^{-1}S) = S^\top(T + W(f_1)^{-1}S). \tag{11}$$

---

[4]Since $r_{j-1} \approx \hat{y}(f_i) - m - Kv_{j-1} = \hat{y}(f_i) - m_{i,j-1}(X)$.

---

**Algorithm 3: Recycling: Virtual Solver Run with Optional Compression.**

---

**Input:** Buffers $\boldsymbol{S}, \boldsymbol{T} \in \mathbb{R}^{NC \times B}$, access to products with $\boldsymbol{W}^{-1}$, compression parameter $R \leqslant B$ (optional)
**Output:** $\boldsymbol{C}_0$, updated buffers $\boldsymbol{S}, \boldsymbol{T}$

| | | | **Time** | **Memory** |
|---|---|---|---|---|
| 1 | **procedure** VIRTUALSOLVERRUN$(\boldsymbol{S}, \boldsymbol{T}, \boldsymbol{W}^{-1})$ | | | |
| 2 | $\quad \boldsymbol{M} \leftarrow \boldsymbol{S}^\top(\boldsymbol{T} + \boldsymbol{W}^{-1}\boldsymbol{S})$ | $M = \boldsymbol{S}^\top(\boldsymbol{K} + \boldsymbol{W}^{-1})\boldsymbol{S} \in \mathbb{R}^{B \times B}$ | $\mathcal{O}(B\tau_{\boldsymbol{W}^{-1}} + B^2 NC)$ | $\mathcal{O}(B^2)$ |
| 3 | $\quad \boldsymbol{U}, \boldsymbol{\Lambda} \leftarrow \mathrm{ED}(\boldsymbol{M})$, | Eigendecomposition $\boldsymbol{M} = \boldsymbol{U}\boldsymbol{\Lambda}\boldsymbol{U}^\top$ | $\mathcal{O}(B^3)$ | $\mathcal{O}(B^2)$ |
| | $\quad \boldsymbol{U} = (\boldsymbol{u}_1, \ldots, \boldsymbol{u}_B), \boldsymbol{\Lambda} = \mathrm{diag}(\lambda_1, \ldots, \lambda_B) \in \mathbb{R}^{B \times B}, \lambda_1 \geqslant \ldots \geqslant \lambda_B$ | | | |
| 4 | $\quad$ **procedure** COMPRESSION$(\boldsymbol{U}, \boldsymbol{\Lambda}, R)$ | | | |
| 5 | $\quad\quad \boldsymbol{U} \leftarrow (\boldsymbol{u}_1, \ldots, \boldsymbol{u}_R), \boldsymbol{\Lambda} \leftarrow \mathrm{diag}(\lambda_1, \ldots, \lambda_R)$ | Truncation to $R$ eigenpairs | | $\mathcal{O}(BR)$ |
| 6 | $\quad \boldsymbol{S} \leftarrow \boldsymbol{S}\boldsymbol{U}, \boldsymbol{T} \leftarrow \boldsymbol{T}\boldsymbol{U}$ | Update buffers | $\mathcal{O}(BRNC)$ | $\mathcal{O}(RNC)$ |
| 7 | $\quad \boldsymbol{Q}_0 \leftarrow \boldsymbol{S}\boldsymbol{\Lambda}^{-1/2}$ | Construct root $\boldsymbol{C}_0 = \boldsymbol{Q}_0\boldsymbol{Q}_0^\top = \boldsymbol{S}\boldsymbol{\Lambda}^{-1}\boldsymbol{S}^\top$ | $\mathcal{O}(R^2 NC)$ | $\mathcal{O}(RNC)$ |
| 8 | $\quad$ **return** $\boldsymbol{C}_0 \leftarrow \boldsymbol{Q}_0\boldsymbol{Q}_0^\top$ and $\boldsymbol{S}, \boldsymbol{T}$ | $\boldsymbol{C}_0$ has rank $R$ | | |

$\boldsymbol{C}_0$ is *never* formed explicitly in memory but evaluated lazily via its root $\boldsymbol{Q}_0$, i.e. $\boldsymbol{w} \mapsto \boldsymbol{C}_0\boldsymbol{w} = \boldsymbol{Q}_0(\boldsymbol{Q}_0^\top \boldsymbol{w})$.

---

So, we can *imitate* a solver run with the previous actions $\boldsymbol{S}$ and construct $\boldsymbol{C}$ without ever having to multiply with $\boldsymbol{K}$. The associated computational costs comprise memory for the two buffers $\boldsymbol{S}, \boldsymbol{T}$ as well as the runtime costs for matrix-matrix products in Equation (11) and inverting $\boldsymbol{M}$. This virtual solver run is generally orders of magnitude cheaper than running the solver from scratch with new actions (details in Appendix B.3). Within ITERGP (Algorithm 2), we can use this matrix as an initial estimate $\boldsymbol{C}_0 \leftarrow \boldsymbol{C}$ of the precision matrix. Subsequently, the algorithm can proceed as usual with new actions.

The presented recycling approach can easily be extended to all Newton steps. Whenever $\boldsymbol{K}$ is multiplied with an action vector, the vector itself and the resulting vector are appended to the respective buffers $\boldsymbol{S}$ and $\boldsymbol{T}$. For each Newton step, an initial $\boldsymbol{C}_0$ can be constructed via Algorithm 3.

**Numerical Perspective.** Crucially, the above strategy does not affect the solver's convergence properties: From a numerical linear algebra viewpoint, the strategy above is a form of *subspace recycling* (Parks et al., 2006). Specifically, $\boldsymbol{C}_0$, as described above, defines a *deflation preconditioner* (Frank & Vuik, 2001): The projection of the initial residual $\boldsymbol{r}_0 = (\hat{\boldsymbol{y}} - \boldsymbol{m}) - \hat{\boldsymbol{K}}\boldsymbol{v}_0$ for the first iterate $\boldsymbol{v}_0 = \boldsymbol{C}_0(\hat{\boldsymbol{y}} - \boldsymbol{m})$ onto the subspace $\mathrm{span}\{\boldsymbol{S}\}$ spanned by the actions is zero (see Appendix A.4 for details). That means, the solution within the subspace $\mathrm{span}\{\boldsymbol{S}\}$ is already perfectly identified at initialization.

**Probabilistic Perspective.** Via Equation (9), we can quantify the effect of $\boldsymbol{C}_0$ on the total marginal uncertainty of predictions at the training data $\mathrm{Tr}(K_{i,0}(\boldsymbol{X}, \boldsymbol{X})) = \mathrm{Tr}(\boldsymbol{K}) - \mathrm{Tr}(\boldsymbol{K}\boldsymbol{C}_0\boldsymbol{K})$. Assuming observation noise $\boldsymbol{W}^{-1} = \boldsymbol{0}$ and all actions in $\boldsymbol{S}$ eigenvectors of $\hat{\boldsymbol{K}} = \boldsymbol{K}$, it simplifies to

$$\mathrm{Tr}(K_{i,0}(\boldsymbol{X}, \boldsymbol{X})) = \mathrm{Tr}(\boldsymbol{K}) - \mathrm{Tr}(\boldsymbol{M}), \tag{12}$$

see Appendix A.4. The second term $\mathrm{Tr}(\boldsymbol{M})$ describes the *reduction* of the prior uncertainty due to $\boldsymbol{C}_0$. It can be maximized (which is our goal) when $\boldsymbol{S}$ contains those eigenvectors of $\hat{\boldsymbol{K}}$ with the largest eigenvalues. We take this insight as motivation for a buffer compression approach that we describe next.

### 3.4 Compression: Memory-Efficient Beliefs

Whenever $\boldsymbol{K}$ is applied to an action vector, the buffers $\boldsymbol{S}, \boldsymbol{T} \in \mathbb{R}^{NC \times B}$ grow by $NC$ entries. To limit memory requirements for large-scale data, we propose a compression strategy (see Algorithm 3).

**Compression via Truncation.** In Algorithm 3, $\boldsymbol{M}^{-1} \in \mathbb{R}^{B \times B}$ is computed via an eigendecomposition $\boldsymbol{M} = \boldsymbol{U}\boldsymbol{\Lambda}\boldsymbol{U}$, such that $\boldsymbol{C}_0 = \boldsymbol{Q}_0\boldsymbol{Q}_0^\top$ can be represented via its matrix root $\boldsymbol{Q}_0 := \boldsymbol{S}\boldsymbol{U}\boldsymbol{\Lambda}^{-1/2}$ for efficient storage and matrix-vector multiplies. To limit memory usage, we can use a *truncated* eigendecomposition of $\boldsymbol{M}$. Based on the intuition we gained from Equation (12), it makes sense to keep the *largest* eigenvalues (to maximize the trace) and corresponding eigenvectors. Keeping the $R$ largest eigenvalues/-vectors yields a rank $R$ approximation $\tilde{\boldsymbol{M}} = \tilde{\boldsymbol{U}}\tilde{\boldsymbol{\Lambda}}\tilde{\boldsymbol{U}}$ of $\boldsymbol{M}$.

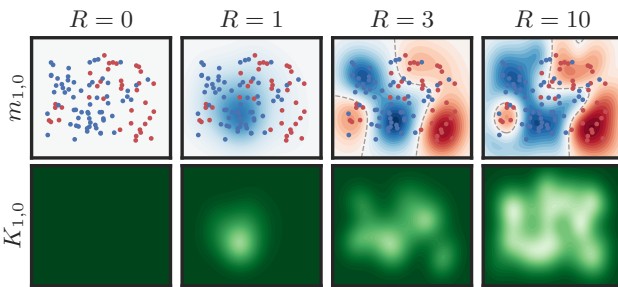

**Figure 4: Compressed Beliefs.** Recycled initial beliefs in the *second* Newton step ($i = 1$) with means $m_{1,0}$ *(Top)* and (co-)variance functions $K_{1,0}$ *(Bottom)* using compression with different buffer sizes $R \in \{0, 1, 3, 10\}$. Buffer size $R = 0$ is equivalent to not using recycling. The larger the buffer size/rank of $C_0$, the more expressive the belief. Details in Appendix C.1.

**Compression as Re-Weighting Actions.** Forming $C_0 = S\tilde{M}^{-1}S^\top$ from the above approximation is equivalent to a virtual solver run with the modified buffers $\tilde{S} = SU \in \mathbb{R}^{NC \times R}$, $\tilde{T} = K(SU) = TU \in \mathbb{R}^{NC \times R}$ in Equation (11). This shows that the truncated eigendecomposition effectively re-weights the previous $B$ actions to form $R$ new ones—and the weights are the eigenvectors from $M$ that maximize the uncertainty reduction. The limit $R$ on the buffer size controls the *memory usage* as well as the rank of $C_0$ and thereby the *expressiveness* of the associated belief (see Figure 4).

### 3.5 Cost Analysis of IterNCGP

IterNCGP's total runtime is dominated by the repeated application of $K$ in Algorithm 2, i.e. $\mathcal{O}(J\tau_K)$, with $J$ describing the *total* number of solver iterations over *all* Newton steps. $\tau_K$ denotes the cost of a single matrix-vector product with $K$. Typically, $\tau_K$ is quadratic in the number of training data points. In terms of memory, the buffers $S$, $T$ and the matrix root $Q$ are the decisive factors with $\mathcal{O}(BNC)$. Without compression, their final size is $B = J$. Otherwise, their maximum size is given by the sum of the rank bound $R$ and the maximum solver iterations in Algorithm 2 (Appendix B.3 provides an in-depth discussion of runtime and memory costs).

## 4 Related Work

The Laplace approximation (Bishop, 2006; MacKay, 1992; Rue et al., 2009; Spiegelhalter & Lauritzen, 1990) is commonly used for approximate inference in (Bayesian) Generalized Linear Models. Here, we consider the function-space generalization of Bayesian Generalized Linear Models, namely non-conjugate GPs, for which a multitude of approximate methods have been proposed, arguably the most popular being variational approaches (e.g. Khan et al., 2012), such as SVGP (Hensman et al., 2015; Titsias, 2009). In contrast, to address the computational shortcomings of NCGPs on large datasets, we leverage iterative methods to obtain and efficiently update low-rank approximations. Similar approaches were used previously to accelerate the conjugate Gaussian special case (Cunningham et al., 2008; Gardner et al., 2018; Guhaniyogi & Dunson, 2015; Murray, 2009; Wang et al., 2019; Wenger et al., 2022a), binary classification (Zhang et al., 2014) and general Bayesian linear inverse problems (Spantini et al., 2015). Trippe et al. (2019) is closest in spirit to our approach if viewed from a weight-space perspective. Their choice of low-rank projection corresponds to a specific policy in our framework. Our approach not only enables the use of policies that are more suited to the given link function, but also saves additional computation, as well as memory, via recycling and compression. In each Newton iteration, the posterior for the current regression problem is approximated via IterGP (Wenger et al., 2022b), which internally uses a probabilistic linear solver (Cockayne et al., 2019a; Hennig, 2015; Wenger & Hennig, 2020). Therefore, IterNCGP is a probabilistic numerical method (Cockayne et al., 2019b; Hennig et al., 2015; 2022; Oates & Sullivan, 2019): It quantifies uncertainty arising from limited computation.

## 5 Experiments

We apply IterNCGP to a Poisson regression problem to explore the trade-off between the number of (outer loop) mode-finding steps and (inner loop) solver iterations (Section 5.1). In Section 5.2, we demonstrate our algorithm's scalability and the impact of compression on performance.

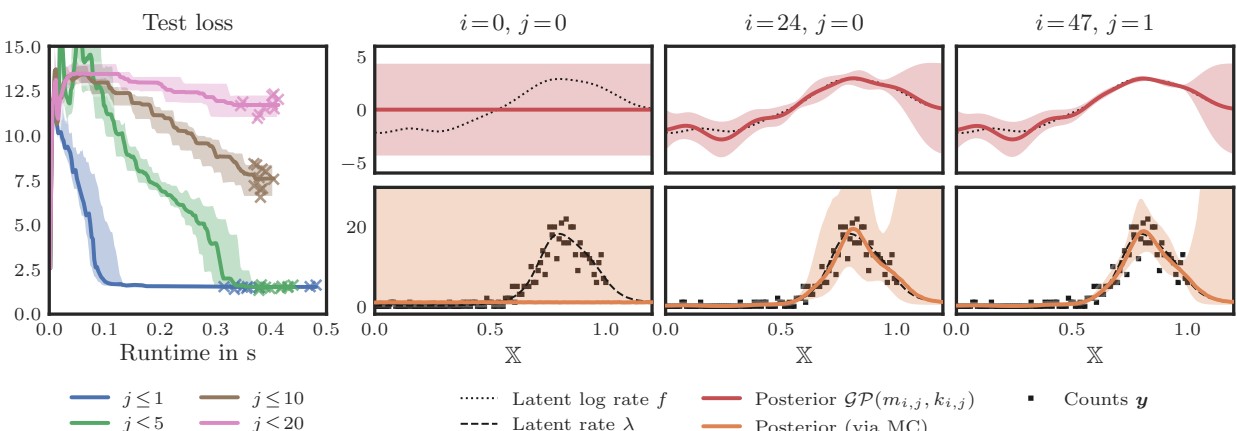

**Figure 5: Poisson Regression with IterNCGP.** *(Left)* Test loss performance for IterNCGP-CG with recycling and four schedules ($j \leq 1, 5, 10$ or $20$) over $100, 20, 10$ or $5$ steps (always using the same total budget of 100 iterations). For each schedule, the median (solid line) and min/max (shaded area) over 10 runs are reported. The crosses indicate the end of each run. *(Right)* Posterior $\mathcal{GP}\,(m_{i,j}, k_{i,j})$ for the latent log rate $f$ *(Top)* and the corresponding belief about the rate $\lambda$ *(Bottom)* computed via MC at three timepoints during a run of IterNCGP. The shaded 95% credible intervals show how stopping early trades less computation for increased uncertainty. Details in Appendix C.2.

## 5.1 Poisson Regression

Consider count data $\boldsymbol{y} \in \mathbb{N}_0^N$ generated from a Poisson likelihood with unknown rate $\lambda \colon \mathbb{X} \to \mathbb{R}_+$. The log rate $f$ is modeled by a GP. See Appendix C.2 for details.

**Data & Model.** We generate a synthetic dataset by (i) sampling the log rate from a GP with an RBF kernel (ii) transforming it into the latent rate $\lambda$ by exponentiation, and (iii) sampling counts $y_n \in \mathbb{N}_0$ from the Poisson distribution with rate $\lambda(\boldsymbol{x}_n)$. The functions $f$, $\lambda$, and the resulting count data are shown in Figure 5 *(Right)*. Our model uses the same RBF prior GP to avoid model mismatch.

**Newton Steps vs. Solver Iterations.** From a practical standpoint, the performance achievable within a given budget of solver iterations is highly relevant: How many linear solver iterations should be performed for each regression problem before updating the problem to maximize performance? To investigate this, we use IterNCGP-CG and distribute 100 iterations uniformly over $\{5, 10, 20, 100\}$ outer loop steps. Each run uses recycling without compression and is repeated 10 times.

**Results.** Figure 5 *(Left)* indicates that the strategy with a single iteration per step is the most efficient. An explanation might be that there is no reason to spend compute on an "outdated" regression problem that could be updated instead. Of course, this only applies if recycling is used, such that the *effective* number of actions accumulates. As long as $B \ll N$, the cost due to repeated recycling ($\mathcal{O}(N)$) is dwarfed by the cost of products with $\boldsymbol{K}$ ($\mathcal{O}(N^2)$). Figure 5 *(Right)* shows an IterNCGP-CG run with one iteration per step. As we spend more computational resources, our estimates approach the underlying latent function and, where data is available, the uncertainty contracts.

## 5.2 Large-Scale GP Multi-Class Classification

Here, we showcase IterNCGP's scalability. See Appendix C.3 for details.

**Data & Model.** We generate $N = 10^5$ data points from a Gaussian mixture model with $C = 10$ classes. We use the softmax likelihood and assume independent GPs (each equipped with a Matérn$(\frac{3}{2})$ kernel) for the $C$ outputs of the latent function. While this experiment uses synthetic data, the latent function is *not* drawn from the assumed GP model and thus the kernel is *not* perfectly identified. Also note that, if we formed $\hat{\boldsymbol{K}}$ in (working) memory explicitly, this would require $(NC)^2 \cdot 8$ byte $= 8000$ GB (in double precision). Solving

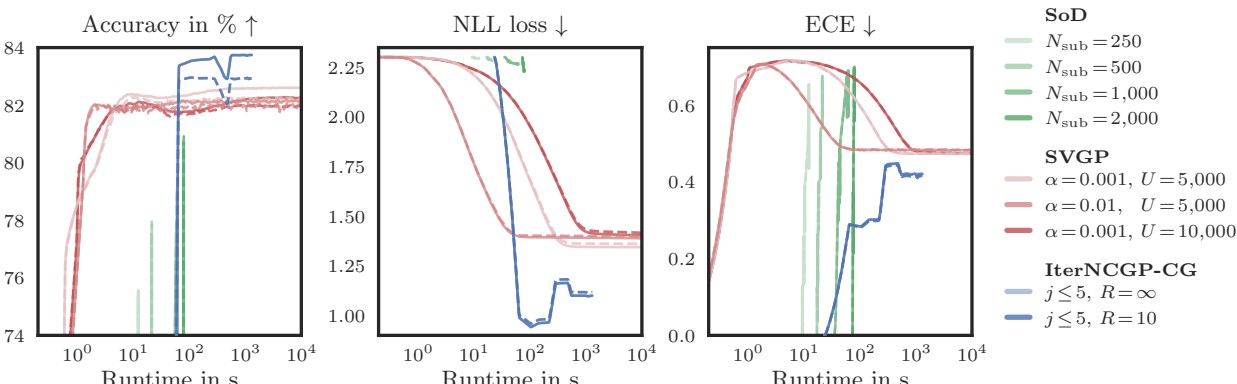

**Figure 6: Large-Scale GP Classification.** Comparison of SoD, SVGP with learning rate $\alpha \in \{0.001, 0.01, 0.05\}$ and $U \in \{1000, 2500, 5000, 10000\}$ inducing points (showing only the best three runs) and IterNCGP-CG with $R \in \{\infty, 10\}$ on a classification problem with $N = 10^5$ training points and $C = 10$ classes in terms of accuracy *(Left)*, NLL loss *(Center)* and ECE *(Right)*. Performance metrics are averaged over five runs and are shown as solid (training set) or dashed (test set) lines. IterNCGP-CG performs best in *all* three performance metrics, with minimal memory requirements. The two variants with $R \in \{\infty, 10\}$ are visually indistinguishable, i.e. the performance is not affected by compression. Details in Appendix C.3.

the linear systems *precisely*, e.g. via Cholesky decomposition, is therefore infeasible, whereas our family of methods is matrix-free and can still be applied.

**Methods.** We compare the subset of data (SoD) approach from Section 3.2, the popular variational approximation SVGP (Hensman et al., 2015; Titsias, 2009) and our IterNCGP-CG. For SoD, we materialize $\hat{K}$ in memory and compute its Cholesky decomposition. Four different subset sizes are used—the largest one $N_{\text{sub}} = 2000$ requires 3.2 GB of memory for $\hat{K}$. For SVGP, we use the implementation provided by GPyTorch (Gardner et al., 2018). We optimize the ELBO for $10^4$ seconds using Adam with batch size 1024 and determine suitable hyperparameters via grid search over the learning rate $\alpha$ and the number of inducing points $U$. Only the best three settings are included in our final benchmark. IterNCGP-CG is applied to the *full* training set with recycling and $R \in \{\infty, 10\}$. The number of solver iterations is limited by $j \leq 5$. We use KeOps (Charlier et al., 2021) and GPyTorch for fast kernel-matrix multiplies. For this work, we consider kernel hyperparameter optimization out of scope—all methods therefore use the same fixed hyperparameters. The benchmark is run on an NVIDIA A100 GPU.

**Results.** Figure 6 shows the average performance of each method over five runs that use different random seeds. Once the matrix is formed in memory, the SoD approaches are very fast—even with $N_{\text{sub}} = 2000$, they converge within 100 s (all SoD runs require only two Newton steps). With increasing $N_{\text{sub}}$, the runs reach higher accuracy at the cost of increased memory requirements. To achieve top performance, SVGP requires a large number of inducing points (at the cost of slower training). Increasing the learning rate to compensate results in instabilities—these runs do not exhibit competitive performance. Both SoD and SVGP fall short of IterNCGP-CG in all three performance metrics. Using recycling, IterNCGP-CG maintains low loss/high accuracy throughout training, even when compression is used. It reaches the lowest final negative log-likelihood (NLL) and expected calibration error (ECE) demonstrating better uncertainty quantification. It is more memory-efficient than SoD (especially with compression), and, in contrast to SVGP, does not require extensive tuning.

**Extension to MNIST.** To demonstrate IterNCGP's applicability to real-world data, we perform a similar experiment on MNIST (Lecun et al., 1998), see Appendix C.4 for details. As KeOps scales poorly with the data dimension ($D = 28^2 = 784$ for MNIST), we revert to GPyTorch's standard kernel implementation, which requires more memory. We thus limit the training data to $N_{\text{sub}} = 20,000$ images. The results (Figure 7) are mostly aligned with Figure 6: IterNCGP-CG outperforms the well-tuned SVGP baselines in terms of accuracy and NLL loss. Only the ECE of IterNCGP is slightly worse than for SVGP. This is easily

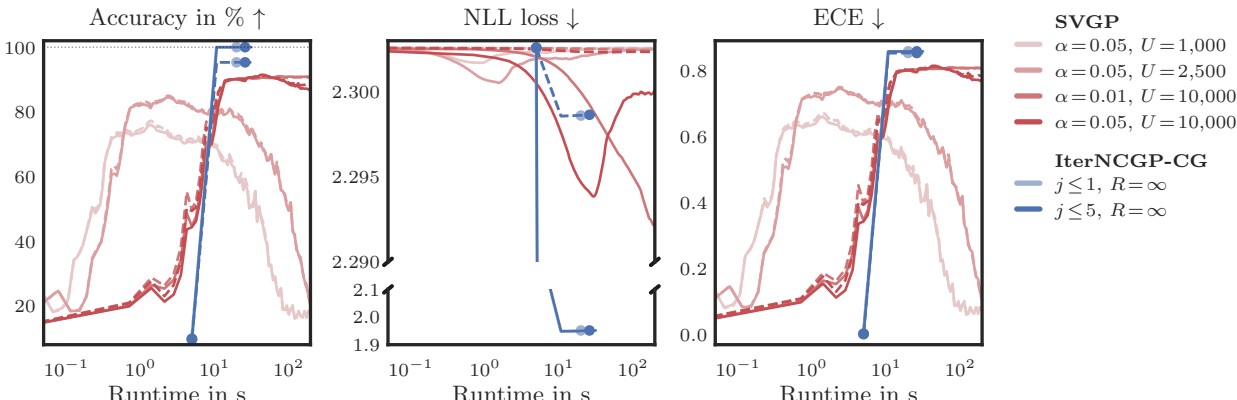

**Figure 7: GP Classification on MNIST.** Comparison of SVGP with learning rate $\alpha \in \{0.001, 0.01, 0.05\}$ and $U \in \{1000, 2500, 5000, 10000\}$ inducing points (showing only the best four runs) and ITERNCGP-CG on a classification problem with $N = 20,000$ training points and $C = 10$ classes in terms of accuracy *(Left)*, NLL loss *(Center)* and ECE *(Right)*. Performance metrics are shown as solid (training set) or dashed (test set) lines. For ITERNCGP, the dots mark the start of a new outer-loop iteration. ITERNCGP-CG performs best in terms of accuracy and NLL loss but slightly worse in terms of ECE. Details in Appendix C.4.

explained, by the fact that one can achieve smaller ECE by accepting lower accuracy—the canonical example being a random baseline, which is perfectly calibrated.

## 6 Conclusion

Non-conjugate Gaussian processes (NCGPs) provide a flexible probabilistic framework encompassing, among others, GP classification and Poisson regression. Training NCGPs on large datasets, however, necessitates approximations. Our method ITERNCGP quantifies and continuously propagates the errors caused by these approximations, in the form of uncertainty. The information collected during training is efficiently recycled and compressed, reducing runtime and memory requirements.

**Limitations.** A limitation of our method is directly inherited from the Laplace approximation: The inherent error in approximating the posterior with a Gaussian is not captured and generally depends on the choice of likelihood. However, under mild regularity conditions, the (relative) error of the Laplace approximation scales inversely proportional to the number of data (Bilodeau et al., 2022; Kass et al., 1990). Since we are considering the large data regime in this work one might reasonably expect the error contribution of the Laplace approximation itself to be small, relative to the error contribution from approximating the MAP via Newton's method, equivalently the error contribution of approximate GP regression.

Another limitation is the lack of a ready-to-use approach for kernel hyperparameter estimation. However, our method is entirely composed of basic linear algebra operations (see Algorithms 1 to 3). Therefore one can in principle simply differentiate with respect to the kernel hyperparameters through the entire solve. Recent work by Wenger et al. (2024) proposes an ELBO objective to perform model selection for ITERGP, which one could therefore readily apply in our setting. We leave the open question on how to optimally choose the number of Newton and ITERGP steps per hyperparameter optimization step for future work.

**Future Work.** So far, we have only explored the policy design space in a limited fashion. The policy controls which areas of the data space are targeted and accounted for in the posterior. Tailoring the actions to the *specific* problem could further increase our method's efficiency. For classification problems, a good strategy might be not to spend compute on data points where the prediction is already definitive.

Finally, a promising application for ITERNCGP may be Bayesian deep learning. A popular approach to equip a neural net with uncertainty is via a Laplace approximation (Khan et al., 2019; MacKay, 1991; Ritter et al., 2018), which is equivalent to a GP classification problem with a neural tangent kernel prior (Immer

et al., 2021; Jacot et al., 2018). There, the SoD approach is regularly used (Immer et al., 2021, Sec. A2.2), for which our approach might offer significant improvements.

## Acknowledgements

The authors gratefully acknowledge co-funding by the European Union (ERC, ANUBIS, 101123955). Views and opinions expressed are however those of the author(s) only and do not necessarily reflect those of the European Union or the European Research Council. Neither the European Union nor the granting authority can be held responsible for them. Philipp Hennig is a member of the Machine Learning Cluster of Excellence, funded by the Deutsche Forschungsgemeinschaft (DFG, German Research Foundation) under Germany's Excellence Strategy - EXC number 2064/1 - Project number 390727645; The authors further gratefully acknowledge financial support by the DFG through Project HE 7114/5-1 in SPP2298/1; the German Federal Ministry of Education and Research (BMBF) through the Tübingen AI Center (FKZ:01IS18039A); and funds from the Ministry of Science, Research and Arts of the State of Baden-Württemberg. Frank Schneider is supported by funds from the Cyber Valley Research Fund. Lukas Tatzel is grateful to the International Max Planck Research School for Intelligent Systems (IMPRS-IS) for support. Jonathan Wenger was supported by the Gatsby Charitable Foundation (GAT3708), the Simons Foundation (542963), the NSF AI Institute for Artificial and Natural Intelligence (ARNI: NSF DBI 2229929) and the Kavli Foundation.

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

# Supplementary Materials

The supplementary materials contain derivations for our theoretical framework and proofs for the mathematical statements in the main text. We also provide implementation specifics and describe our experimental setup in more detail.

# A    Mathematical Details

## A.1    Newton's Method as Sequential GP Regression

In Section 3.1, we reinterpret the Newton iteration as a sequence of GP regression problems. More specifically, we rewrite the posterior predictive mean (Equation (4)) as a GP posterior for a regression problem (Equation (6)). Here, we provide a proof for this connection.

**Proposition A.1** (Reformulation of the Newton Step)
*Let $\boldsymbol{W}(\boldsymbol{f}_i)$ be invertible. Using the transform $\boldsymbol{g} := \boldsymbol{f} - \boldsymbol{m}$ and consequently $\boldsymbol{g}_i = \boldsymbol{f}_i - \boldsymbol{m}$, the Newton step (Equation (3)) can be written as*

$$\boldsymbol{g}_{i+1} = \boldsymbol{K}(\boldsymbol{K} + \boldsymbol{W}(\boldsymbol{f}_i)^{-1})^{-1}\left(\boldsymbol{g}_i + \boldsymbol{W}(\boldsymbol{f}_i)^{-1}\nabla \log p(\boldsymbol{y} \mid \boldsymbol{f}_i)\right).$$

*Proof.* Recall from Equation (3) that

$$\boldsymbol{f}_{i+1} = \boldsymbol{f}_i - \nabla^2\Psi(\boldsymbol{f}_i)^{-1} \cdot \nabla\Psi(\boldsymbol{f}_i), \quad \text{with} \quad \begin{aligned}\nabla\Psi(\boldsymbol{f}_i) &= \nabla \log p(\boldsymbol{y} \mid \boldsymbol{f}_i) - \boldsymbol{K}^{-1}(\boldsymbol{f}_i - \boldsymbol{m})\\ \nabla^2\Psi(\boldsymbol{f}_i) &= -\boldsymbol{W}(\boldsymbol{f}_i) - \boldsymbol{K}^{-1},\end{aligned}$$

where $\boldsymbol{W}(\boldsymbol{f}_i) = -\nabla^2 \log p(\boldsymbol{y} \mid \boldsymbol{f}_i)$ denotes the negative Hessian (with respect to $\boldsymbol{f}$) of the log likelihood evaluated at $\boldsymbol{f}_i$. It holds

$$\begin{aligned}\boldsymbol{f}_{i+1} &= \boldsymbol{f}_i - \nabla^2\Psi(\boldsymbol{f}_i)^{-1} \cdot \nabla\Psi(\boldsymbol{f}_i)\\ &= \boldsymbol{f}_i + (\boldsymbol{W}(\boldsymbol{f}_i) + \boldsymbol{K}^{-1})^{-1} \cdot \left(\nabla \log p(\boldsymbol{y} \mid \boldsymbol{f}_i) - \boldsymbol{K}^{-1}(\boldsymbol{f}_i - \boldsymbol{m})\right)\end{aligned}$$

By substracting $\boldsymbol{m}$ from both sides we obtain

$$\begin{aligned}\boldsymbol{g}_{i+1} &= \boldsymbol{g}_i + (\boldsymbol{W}(\boldsymbol{f}_i) + \boldsymbol{K}^{-1})^{-1} \cdot \left(\nabla \log p(\boldsymbol{y} \mid \boldsymbol{f}_i) - \boldsymbol{K}^{-1}\boldsymbol{g}_i\right)\\ &= (\boldsymbol{W}(\boldsymbol{f}_i) + \boldsymbol{K}^{-1})^{-1}\left((\boldsymbol{W}(\boldsymbol{f}_i) + \boldsymbol{K}^{-1})\boldsymbol{g}_i + \nabla \log p(\boldsymbol{y} \mid \boldsymbol{f}_i) - \boldsymbol{K}^{-1}\boldsymbol{g}_i\right)\\ &= (\boldsymbol{W}(\boldsymbol{f}_i) + \boldsymbol{K}^{-1})^{-1}\left(\boldsymbol{W}(\boldsymbol{f}_i)\boldsymbol{g}_i + \nabla \log p(\boldsymbol{y} \mid \boldsymbol{f}_i)\right)\\ &= (\boldsymbol{W}(\boldsymbol{f}_i) + \boldsymbol{K}^{-1})^{-1}\boldsymbol{W}(\boldsymbol{f}_i)\left(\boldsymbol{g}_i + \boldsymbol{W}(\boldsymbol{f}_i)^{-1}\nabla \log p(\boldsymbol{y} \mid \boldsymbol{f}_i)\right)\\ &= (\boldsymbol{I} + \boldsymbol{W}(\boldsymbol{f}_i)^{-1}\boldsymbol{K}^{-1})^{-1}\left(\boldsymbol{g}_i + \boldsymbol{W}(\boldsymbol{f}_i)^{-1}\nabla \log p(\boldsymbol{y} \mid \boldsymbol{f}_i)\right)\\ &= (\boldsymbol{K}\boldsymbol{K}^{-1} + \boldsymbol{W}(\boldsymbol{f}_i)^{-1}\boldsymbol{K}^{-1})^{-1}\left(\boldsymbol{g}_i + \boldsymbol{W}(\boldsymbol{f}_i)^{-1}\nabla \log p(\boldsymbol{y} \mid \boldsymbol{f}_i)\right)\\ &= \boldsymbol{K}(\boldsymbol{K} + \boldsymbol{W}(\boldsymbol{f}_i)^{-1})^{-1}\left(\boldsymbol{g}_i + \boldsymbol{W}(\boldsymbol{f}_i)^{-1}\nabla \log p(\boldsymbol{y} \mid \boldsymbol{f}_i)\right)\\ &= \boldsymbol{K}\hat{\boldsymbol{K}}(\boldsymbol{f}_i)^{-1}\left(\boldsymbol{g}_i + \boldsymbol{W}(\boldsymbol{f}_i)^{-1}\nabla \log p(\boldsymbol{y} \mid \boldsymbol{f}_i)\right),\end{aligned}$$

with $\hat{\boldsymbol{K}}(\boldsymbol{f}_i) := \boldsymbol{K} + \boldsymbol{W}(\boldsymbol{f}_i)^{-1}$. $\qquad\square$

**Newton's Method as Sequential GP Regression.** Using the LA at $\boldsymbol{f}_i$, we obtain a GP posterior (see Equations (4) and (5) in Section 2). With Proposition A.1 (i.e. assuming that $\boldsymbol{W}(\boldsymbol{f}_i)^{-1}$ exists), we can rewrite Equation (4) as

$$\begin{aligned}m_{i,*}(\cdot) &= m(\cdot) + K(\cdot, \boldsymbol{X})\boldsymbol{K}^{-1}(\boldsymbol{f}_{i+1} - \boldsymbol{m})\\ &= m(\cdot) + K(\cdot, \boldsymbol{X})\boldsymbol{K}^{-1}\boldsymbol{g}_{i+1}\\ &= m(\cdot) + K(\cdot, \boldsymbol{X})\boldsymbol{K}^{-1}\boldsymbol{K}\hat{\boldsymbol{K}}(\boldsymbol{f}_i)^{-1}\left(\boldsymbol{g}_i + \boldsymbol{W}(\boldsymbol{f}_i)^{-1}\nabla \log p(\boldsymbol{y} \mid \boldsymbol{f}_i)\right)\\ &= m(\cdot) + K(\cdot, \boldsymbol{X})\hat{\boldsymbol{K}}(\boldsymbol{f}_i)^{-1}\left(\boldsymbol{f}_i + \boldsymbol{W}(\boldsymbol{f}_i)^{-1}\nabla \log p(\boldsymbol{y} \mid \boldsymbol{f}_i) - \boldsymbol{m}\right)\\ &= m(\cdot) + K(\cdot, \boldsymbol{X})\hat{\boldsymbol{K}}(\boldsymbol{f}_i)^{-1}(\hat{\boldsymbol{y}}(\boldsymbol{f}_i) - \boldsymbol{m}),\end{aligned}$$

where $\hat{\boldsymbol{y}}(\boldsymbol{f}_i) := \boldsymbol{f}_i + \boldsymbol{W}(\boldsymbol{f}_i)^{-1}\nabla \log p(\boldsymbol{y} \mid \boldsymbol{f}_i)$. This proves Equation (6). Together with Equation (5), $m_{i,*}$ defines a GP posterior for a GP regression problem with pseudo targets $\hat{\boldsymbol{y}}(\boldsymbol{f}_i)$ observed with Gaussian noise $\mathcal{N}\left(\boldsymbol{0}, \boldsymbol{W}(\boldsymbol{f}_i)^{-1}\right)$ (Rasmussen & Williams, 2006, Eqs. (2.24) and (2.38)).

Equation (6) requires solving the linear system $\hat{\boldsymbol{K}}(\boldsymbol{f}_i) \cdot \boldsymbol{v} = \hat{\boldsymbol{y}}(\boldsymbol{f}_i) - \boldsymbol{m}$ of size $NC$. Then, $m_{i,*}(\cdot) = m(\cdot) + K(\cdot, \boldsymbol{X})\boldsymbol{v}$. In Proposition A.1, we can write $\boldsymbol{g}_{i+1}$ as $\boldsymbol{g}_{i+1} = \boldsymbol{K}\boldsymbol{v}$, i.e. $\boldsymbol{f}_{i+1} = \boldsymbol{K}\boldsymbol{v} + \boldsymbol{m}$. So, *both* the predictive mean $m_{i,*}$ *and* the Newton update $\boldsymbol{f}_{i+1}$ follow directly from the solution $\boldsymbol{v}$. In that sense, performing inference and computing Newton iterates are equivalent.

**What If $\boldsymbol{W}(\boldsymbol{f}_i)$ Is *Not* Invertible?** For multi-class classification, $\boldsymbol{W}$ has rank $N(C-1)$ and thus $\boldsymbol{W}^{-1}$ does not exist. Therefore, we use its pseudo-inverse $\boldsymbol{W}^\dagger$ instead. We derive an explicit expression for $\boldsymbol{W}^\dagger$ in Appendix A.6 which allows for efficient matrix-vector multiplies.

## A.2 Our Algorithm is an Extension of IterGP

Our algorithm IterNCGP uses IterGP as a core building block. IterNCGP's outer loop (Algorithm 1) can be understood as a sequence of GP regression problems and we use IterGP (that implements the inner loop, see Algorithm 2) for finding approximate solutions to each of these problems. In the case of GP regression (i.e. with a Gaussian likelihood), the outer loop collapses to a *single* iteration and IterNCGP coincides exactly with IterGP, as we show in the following.

**Theorem A.2** (Generalization of IterGP)
*For a Gaussian likelihood $p(\boldsymbol{y} \mid \boldsymbol{f}) = \mathcal{N}(\boldsymbol{y}; \boldsymbol{f}, \boldsymbol{\Lambda})$, IterNCGP converges in a single Newton step (i.e. $\boldsymbol{f}_1 = \boldsymbol{f}_{MAP}$) and IterNCGP (Algorithm 1) coincides exactly with IterGP (Algorithm 2).*

*Proof.* Since the likelihood is Gaussian $p(\boldsymbol{y} \mid \boldsymbol{f}) = \mathcal{N}(\boldsymbol{y}; \boldsymbol{f}, \boldsymbol{\Lambda})$, the log likelihood is given by

$$\log p(\boldsymbol{y} \mid \boldsymbol{f}) \overset{c}{=} -\frac{1}{2}(\boldsymbol{f} - \boldsymbol{y})^\top \boldsymbol{\Lambda}^{-1}(\boldsymbol{f} - \boldsymbol{y}).$$

This gives rise to a log-posterior (Equation (2))

$$
\begin{aligned}
\Psi(\boldsymbol{f}) &:= \log p(\boldsymbol{f} \mid \boldsymbol{X}, \boldsymbol{y}) \\
&\overset{c}{=} \log p(\boldsymbol{y} \mid \boldsymbol{f}) - \frac{1}{2}(\boldsymbol{f} - \boldsymbol{m})^\top \boldsymbol{K}^{-1}(\boldsymbol{f} - \boldsymbol{m}) \\
&= -\frac{1}{2}(\boldsymbol{f} - \boldsymbol{y})^\top \boldsymbol{\Lambda}^{-1}(\boldsymbol{f} - \boldsymbol{y}) - \frac{1}{2}(\boldsymbol{f} - \boldsymbol{m})^\top \boldsymbol{K}^{-1}(\boldsymbol{f} - \boldsymbol{m})
\end{aligned}
$$

that is *quadratic* in $\boldsymbol{f}$. The first Newton iterate $\boldsymbol{f}_1$ therefore coincides with log-posterior's maximizer $\boldsymbol{f}_1 = \boldsymbol{f}_{\mathrm{MAP}}$. The outer loop of IterNCGP thus reduces to a single iteration.

How does this step look from the perspective of the IterNCGP algorithm? First note that $\boldsymbol{W}(\boldsymbol{f}) = -\nabla^2 \log p(\boldsymbol{y} \mid \boldsymbol{f}) \equiv \boldsymbol{\Lambda}^{-1}$. Given an initial $\boldsymbol{f}_0$, IterNCGP computes the observation noise $\boldsymbol{W}^{-1}(\boldsymbol{f}_0) = \boldsymbol{\Lambda}$ and pseudo regression targets $\hat{\boldsymbol{y}}(\boldsymbol{f}_0) = \boldsymbol{f}_0 + \boldsymbol{W}(\boldsymbol{f}_0)^{-1}\nabla \log p(\boldsymbol{y} \mid \boldsymbol{f}_0) = \boldsymbol{f}_0 - \boldsymbol{\Lambda}\boldsymbol{\Lambda}^{-1}(\boldsymbol{f}_0 - \boldsymbol{y}) = \boldsymbol{y}$. Both these quantities are *independent* of the initialization $\boldsymbol{f}_0$. Thus, the first (and only) regression problem that IterNCGP forms in the outer loop is the *original* regression problem (defined by labels $\boldsymbol{y}$ and the observation noise $\Lambda$) and IterGP is applied to solve that regression problem. This shows that our framework recovers IterGP in the case of a Gaussian likelihood and our algorithm can thus be regarded as an extension thereof. $\square$

## A.3 The Marginal Uncertainty Decreases in the Inner Loop

We claim in Section 3.1 that the marginal uncertainty captured by $K_{i,j}(\boldsymbol{x}, \boldsymbol{x}) \in \mathbb{R}^{C \times C}$ (see Equation (9)) within a Newton step decreases with each solver iteration $j$. Here, we provide the proof for this statement.

**Proposition A.3** (The Uncertainty Decreases in the Inner Loop)
*For each $i$ it holds (element-wise) that $\mathrm{diag}(K_{i,j}(\boldsymbol{x}, \boldsymbol{x})) \geqslant \mathrm{diag}(K_{i,k}(\boldsymbol{x}, \boldsymbol{x}))$ for any $k \geqslant j$ and arbitrary $\boldsymbol{x}$.*

*Proof.* To see this, we rewrite $\boldsymbol{C}_j$ as a sum of $j$ rank-1 matrices $\boldsymbol{C}_j = \sum_{\ell=1}^{j} \boldsymbol{d}_\ell \boldsymbol{d}_\ell^\top$ and substitute this into Equation (9). It holds that

$$
\begin{aligned}
\operatorname{diag}(K_{i,j}(\boldsymbol{x},\boldsymbol{x})) &= \operatorname{diag}(K(\boldsymbol{x},\boldsymbol{x})) - \sum_{\ell=1}^{j} \operatorname{diag}(\underbrace{K(\boldsymbol{x},\boldsymbol{X})\boldsymbol{d}_\ell}_{=:\hat{\boldsymbol{d}}_\ell} \underbrace{\boldsymbol{d}_\ell^\top K(\boldsymbol{X},\boldsymbol{x})}_{=\hat{\boldsymbol{d}}_\ell^\top}) \\
&= \operatorname{diag}(K(\boldsymbol{x},\boldsymbol{x})) - \sum_{\ell=1}^{j} \hat{\boldsymbol{d}}_\ell^2 \qquad\qquad \text{The square is applied element-wise.} \\
&\geqslant \operatorname{diag}(K(\boldsymbol{x},\boldsymbol{x})) - \sum_{\ell=1}^{k} \hat{\boldsymbol{d}}_\ell^2 \quad \text{for } k \geqslant j \\
&= \operatorname{diag}(K_{i,k}(\boldsymbol{x},\boldsymbol{x}))
\end{aligned}
$$

for any $\boldsymbol{x} \in \mathbb{X}$. $\qquad\square$

## A.4 Virtual Solver Run

In Section 3.3, we showed that it is possible to *imitate* a solver run using the *previous* actions on the *new* problem, without ever having to multiply by $\boldsymbol{K}$. The pseudo code is given in Algorithm 3. Here, we discuss the numerical and probabilistic perspective on that procedure in more detail and provide derivations for the statements in the main text.

**Numerical Perspective.** Let $\boldsymbol{S} = (\boldsymbol{s}_1, \dots, \boldsymbol{s}_B)$ the matrix of stacked linearly independent actions. We use $\boldsymbol{C}_0 = \boldsymbol{S}(\boldsymbol{S}^\top \hat{\boldsymbol{K}} \boldsymbol{S})^{-1}\boldsymbol{S}^\top$ (see Equation (10)) as an initial estimate of the precision matrix in Algorithm 2. The corresponding initial residual (see Algorithm 2) $\boldsymbol{r}_0 = (\hat{\boldsymbol{y}} - \boldsymbol{m}) - \hat{\boldsymbol{K}}\boldsymbol{v}_0$ for the first iterate $\boldsymbol{v}_0 = \boldsymbol{C}_0(\hat{\boldsymbol{y}} - \boldsymbol{m})$ can be decomposed into $\boldsymbol{P_S}\boldsymbol{r}_0$ and $(\boldsymbol{I} - \boldsymbol{P_S})\boldsymbol{r}_0$. $\boldsymbol{P_S} = \boldsymbol{S}(\boldsymbol{S}^\top \boldsymbol{S})^{-1}\boldsymbol{S}^\top$ is the orthogonal projection onto the subspace span$\{\boldsymbol{S}\}$ spanned by the actions.

**Proposition A.4** (Residual in span$\{\boldsymbol{S}\}$ Is Zero)
*The orthogonal projection $\boldsymbol{P_S}\boldsymbol{r}_0$ of the initial residual $\boldsymbol{r}_0$ onto* span$\{\boldsymbol{S}\}$ *is zero.*

*Proof.* It holds that

$$
\begin{aligned}
\boldsymbol{P_S}\boldsymbol{r}_0 &= \boldsymbol{P_S}(\hat{\boldsymbol{y}} - \boldsymbol{m}) - \boldsymbol{P_S}\hat{\boldsymbol{K}}\boldsymbol{v}_0 \\
&= \boldsymbol{P_S}(\hat{\boldsymbol{y}} - \boldsymbol{m}) - \boldsymbol{P_S}\hat{\boldsymbol{K}}\boldsymbol{C}_0(\hat{\boldsymbol{y}} - \boldsymbol{m}) \\
&= \boldsymbol{P_S}(\hat{\boldsymbol{y}} - \boldsymbol{m}) - \underbrace{\boldsymbol{S}(\boldsymbol{S}^\top \boldsymbol{S})^{-1}(\boldsymbol{S}^\top}_{=\boldsymbol{P_S}} \hat{\boldsymbol{K}} \underbrace{\boldsymbol{S})(\boldsymbol{S}^\top \hat{\boldsymbol{K}} \boldsymbol{S})^{-1}\boldsymbol{S}^\top}_{=\boldsymbol{C}_0}(\hat{\boldsymbol{y}} - \boldsymbol{m}) \\
&= \boldsymbol{P_S}(\hat{\boldsymbol{y}} - \boldsymbol{m}) - \underbrace{\boldsymbol{S}(\boldsymbol{S}^\top \boldsymbol{S})^{-1}\boldsymbol{S}^\top}_{=\boldsymbol{P_S}}(\hat{\boldsymbol{y}} - \boldsymbol{m}) \\
&= 0. \qquad\qquad\qquad\qquad\qquad\qquad\qquad\qquad\qquad\qquad\qquad\qquad\square
\end{aligned}
$$

Proposition A.4 shows that the residual in span$\{\boldsymbol{S}\}$ is zero. In that sense, the solution within this subspace is already perfectly identified at initialization. The remaining residual thus lies in the orthogonal complement of span$\{\boldsymbol{S}\}$ which can be targeted through additional actions. If we measure the error in the representer weights $\boldsymbol{v} - \boldsymbol{v}_0$, a similar results holds, as we show in the following.

**Proposition A.5** (Error in Representer Weights in span$\{\boldsymbol{S}\}$ Is Zero)
*The $\hat{\boldsymbol{K}}$-orthogonal projection of the representer weights approximation error $\hat{\boldsymbol{P}}_{\boldsymbol{S}}(\boldsymbol{v} - \boldsymbol{v}_0)$ onto* span$\{\boldsymbol{S}\}$ *is zero.*

*Proof.* The $\hat{\boldsymbol{K}}$-orthogonal (orthogonal with respect to the inner product $\langle \cdot, \cdot \rangle_{\hat{\boldsymbol{K}}}$) projection onto the subspace span$\{\boldsymbol{S}\}$ spanned by the actions is given by $\hat{\boldsymbol{P}}_{\boldsymbol{S}} = \boldsymbol{C}_0\hat{\boldsymbol{K}}$ (Wenger et al., 2022b, Section S2.1). It holds that

$$
\hat{\boldsymbol{P}}_{\boldsymbol{S}}(\boldsymbol{v} - \boldsymbol{v}_0) = \boldsymbol{C}_0\hat{\boldsymbol{K}}(\boldsymbol{v} - \boldsymbol{v}_0)
$$

$$= \boldsymbol{C}_0 \hat{\boldsymbol{K}} \hat{\boldsymbol{K}}^{-1}(\hat{\boldsymbol{y}} - \boldsymbol{m}) - \boldsymbol{C}_0 \hat{\boldsymbol{K}} \boldsymbol{C}_0 \underbrace{(\hat{\boldsymbol{y}} - \boldsymbol{m})}_{= \hat{\boldsymbol{K}} \boldsymbol{v}}$$

$$= \boldsymbol{C}_0(\hat{\boldsymbol{y}} - \boldsymbol{m}) - \underbrace{\boldsymbol{C}_0 \hat{\boldsymbol{K}}}_{= \hat{\boldsymbol{P}}_S} \underbrace{\boldsymbol{C}_0 \hat{\boldsymbol{K}}}_{= \hat{\boldsymbol{P}}_S} \boldsymbol{v}$$

$$= \boldsymbol{C}_0(\hat{\boldsymbol{y}} - \boldsymbol{m}) - \boldsymbol{C}_0 \underbrace{\hat{\boldsymbol{K}} \boldsymbol{v}}_{= \hat{\boldsymbol{y}} - \boldsymbol{m}}$$

$$= 0,$$

where we used that $\boldsymbol{v} = \hat{\boldsymbol{K}}^{-1}(\hat{\boldsymbol{y}} - \boldsymbol{m})$ is the solution of the GP regression linear system, $\boldsymbol{v}_0 = \boldsymbol{C}_0(\hat{\boldsymbol{y}} - \boldsymbol{m})$ and the idempotence of the projection matrix $\hat{\boldsymbol{P}}_S = \hat{\boldsymbol{P}}_S \hat{\boldsymbol{P}}_S$. □

**Probabilistic Perspective.** Equation (9) describes the effect of $\boldsymbol{C}_0$ from a probabilistic perspective. Initializing $\boldsymbol{C}_0 = \boldsymbol{0}$ in step $i$ results in $m_{i,0} = m(\cdot)$ (prior mean) and $K_{i,0} = K(\cdot, \cdot)$ (prior covariance) since the *reduction* of uncertainty $K(\cdot, \boldsymbol{X}) \boldsymbol{C}_0 K(\boldsymbol{X}, \cdot)$ is zero. This case, where no information from past steps is included, is illustrated in the first column $R = 0$ in Figure 4.

**Special Case.** We consider a special case, where the general intricate form of the total marginal variance from Section 3.4

$$\operatorname{Tr}(K_{i,0}(\boldsymbol{X}, \boldsymbol{X})) = \operatorname{Tr}(\boldsymbol{K}) - \operatorname{Tr}(\boldsymbol{K} \boldsymbol{C}_0 \boldsymbol{K}) \tag{13}$$

collapses. Let $\lambda_1, \ldots, \lambda_{NC} > 0$ denote the eigenvalues of $\hat{\boldsymbol{K}}$ and $\boldsymbol{b}_1, \ldots, \boldsymbol{b}_{NC}$ the corresponding (pairwise orthogonal) eigenvectors. We make the following two assumptions: **(A1):** We assume $\boldsymbol{W}^{-1} = \boldsymbol{0}$, i.e. $\hat{\boldsymbol{K}} = \boldsymbol{K}$. **(A2):** We assume that the actions coincide with a subset $\mathbb{L} \subseteq \{1, \ldots, NC\}$ of $\hat{\boldsymbol{K}}$'s eigenvectors $\boldsymbol{S} = (\boldsymbol{b}_l)_{l \in \mathbb{L}} \in \mathbb{R}^{NC \times |\mathbb{L}|}$.

**Proposition A.6** (Total Marginal Uncertainty)
*Under assumptions* **(A1)** *and* **(A2)** *it holds that*

$$\operatorname{Tr}(K_{i,0}(\boldsymbol{X}, \boldsymbol{X})) = \operatorname{Tr}(\boldsymbol{K}) - \operatorname{Tr}(\boldsymbol{M}).$$

*Proof.* Let $\boldsymbol{S} = (\boldsymbol{b}_l)_{l \in \mathbb{L}} \in \mathbb{R}^{NC \times |\mathbb{L}|}$ and $\boldsymbol{\Lambda} = \operatorname{diag}((\lambda_l)_{l \in \mathbb{L}}) \in \mathbb{R}^{|\mathbb{L}| \times |\mathbb{L}|}$ contain a subset $\mathbb{L} \subseteq \{1, \ldots, NC\}$ of $\hat{\boldsymbol{K}}$'s eigenpairs. The remaining eigenvectors and eigenvalues are given by $\boldsymbol{S}_+ = (\boldsymbol{b}_l)_{l \notin \mathbb{L}} \in \mathbb{R}^{NC \times NC - |\mathbb{L}|}$ and $\boldsymbol{\Lambda}_+ = \operatorname{diag}((\lambda_l)_{l \notin \mathbb{L}}) \in \mathbb{R}^{NC - |\mathbb{L}| \times NC - |\mathbb{L}|}$. First note that we can write the eigendecomposition of $\hat{\boldsymbol{K}} = \boldsymbol{K}$ as a sum of two components $\hat{\boldsymbol{K}} = \boldsymbol{S} \boldsymbol{\Lambda} \boldsymbol{S}^\top + \boldsymbol{S}_+ \boldsymbol{\Lambda}_+ \boldsymbol{S}_+^\top$, each of which covers one part of the spectrum. It holds

$$\boldsymbol{S}^\top \boldsymbol{S} = \boldsymbol{I}, \quad \boldsymbol{S}_+^\top \boldsymbol{S}_+ = \boldsymbol{I}, \quad \boldsymbol{S}^\top \boldsymbol{S}_+ = \boldsymbol{0} \quad \text{and} \quad \boldsymbol{S}_+^\top \boldsymbol{S} = \boldsymbol{0}$$

since $\hat{\boldsymbol{K}}$ is symmetric and its eigenvectors are thus pairwise orthogonal. It follows

$$\boldsymbol{K} \boldsymbol{S} = (\boldsymbol{S} \boldsymbol{\Lambda} \boldsymbol{S}^\top + \boldsymbol{S}_+ \boldsymbol{\Lambda}_+ \boldsymbol{S}_+^\top) \boldsymbol{S} = \boldsymbol{S} \boldsymbol{\Lambda}$$

$$\boldsymbol{S}^\top \boldsymbol{K} = \boldsymbol{S}^\top (\boldsymbol{S} \boldsymbol{\Lambda} \boldsymbol{S}^\top + \boldsymbol{S}_+ \boldsymbol{\Lambda}_+ \boldsymbol{S}_+^\top) = \boldsymbol{\Lambda} \boldsymbol{S}^\top$$

$$\boldsymbol{M} = \boldsymbol{S}^\top \boldsymbol{K} \boldsymbol{S} = \boldsymbol{S}^\top (\boldsymbol{S} \boldsymbol{\Lambda} \boldsymbol{S}^\top + \boldsymbol{S}_+ \boldsymbol{\Lambda}_+ \boldsymbol{S}_+^\top) \boldsymbol{S} = \boldsymbol{\Lambda}.$$

Plugging those expressions into Equation (13) yields

$$\begin{aligned}
\operatorname{Tr}(K_{i,0}(\boldsymbol{X}, \boldsymbol{X})) &= \operatorname{Tr}(\boldsymbol{K}) - \operatorname{Tr}(\boldsymbol{K} \boldsymbol{C}_0 \boldsymbol{K}) \\
&= \operatorname{Tr}(\boldsymbol{K}) - \operatorname{Tr}(\boldsymbol{K} \boldsymbol{S} \underbrace{(\boldsymbol{S}^\top \hat{\boldsymbol{K}} \boldsymbol{S})^{-1}}_{= \boldsymbol{M}^{-1}} \boldsymbol{S}^\top \boldsymbol{K}) \\
&= \operatorname{Tr}(\boldsymbol{K}) - \operatorname{Tr}(\boldsymbol{S} \boldsymbol{\Lambda} \boldsymbol{\Lambda}^{-1} \boldsymbol{\Lambda} \boldsymbol{S}^\top) \\
&= \operatorname{Tr}(\boldsymbol{K}) - \operatorname{Tr}(\boldsymbol{S}^\top \boldsymbol{S} \boldsymbol{\Lambda}) \\
&= \operatorname{Tr}(\boldsymbol{K}) - \operatorname{Tr}(\boldsymbol{\Lambda}) \\
&= \operatorname{Tr}(\boldsymbol{K}) - \operatorname{Tr}(\boldsymbol{M}) \\
&= \sum_{l \notin \mathbb{L}} \lambda_l.
\end{aligned}$$

The last equation is due to

$$\mathrm{Tr}(\boldsymbol{K}) = \mathrm{Tr}(\boldsymbol{S}\boldsymbol{\Lambda}\boldsymbol{S}^\top) + \mathrm{Tr}(\boldsymbol{S}_+\boldsymbol{\Lambda}_+\boldsymbol{S}_+^\top) = \mathrm{Tr}(\boldsymbol{S}^\top\boldsymbol{S}\boldsymbol{\Lambda}) + \mathrm{Tr}(\boldsymbol{S}_+^\top\boldsymbol{S}_+\boldsymbol{\Lambda}_+) = \mathrm{Tr}(\boldsymbol{\Lambda}) + \mathrm{Tr}(\boldsymbol{\Lambda}_+). \qquad \square$$

Proposition A.6 shows that the *reduction* of the marginal uncertainty is determined by the sum of $\boldsymbol{M}$'s eigenvalues $\sum_{l\in\mathbb{L}} \lambda_l$. If $\boldsymbol{S}$ contains the eigenvectors $\boldsymbol{b}_l$ to the largest eigenvalues (i.e. $\boldsymbol{S}$ is "aligned" with the high-variance subspace of $\hat{\boldsymbol{K}}$), the remaining uncertainty $\sum_{l\notin\mathbb{L}} \lambda_l$ is small. In contrast, if $\boldsymbol{S}$ covers the low-variance subspace of $\hat{\boldsymbol{K}}$, the uncertainty remains largely unaffected.

## A.5 Derivatives of the Poisson Log Likelihood

One of our main experiments in Section 5 is Poisson regression (see Appendix C.2 for details). In order to apply IterNCGP, we have to formulate the problem within the NCGP framework. In particular, we have to specify the derivatives of the log likelihood.

The Poisson likelihood is given by

$$p(\boldsymbol{y} \mid \boldsymbol{f}) = \prod_{n=1}^{N} \frac{\lambda_n^{y_n} \exp(-\lambda_n)}{y_n!},$$

where $y_n \in \mathbb{N}_0$ and $\boldsymbol{\lambda} := \lambda(\boldsymbol{X}) = \exp(f(\boldsymbol{X})) = \exp(\boldsymbol{f})$. Taking the logarithm yields

$$\log p(\boldsymbol{y} \mid \boldsymbol{f}) = \sum_{n=1}^{N} \log\left(\frac{\lambda_n^{y_n} \exp(-\lambda_n)}{y_n!}\right) = \sum_{n=1}^{N} \left(y_n \log(\lambda_n) - \lambda_n - \log(y_n!)\right).$$

The log likelihood's gradient and Hessian with respect to $\boldsymbol{f}$ are therefore given by

$$\nabla \log p(\boldsymbol{y} \mid \boldsymbol{f}) = \boldsymbol{y} - \exp(\boldsymbol{f}) \quad \text{and} \quad \nabla^2 \log p(\boldsymbol{y} \mid \boldsymbol{f}) = -\mathrm{diag}(\exp(\boldsymbol{f})),$$

where the exponential function is applied element-wise. This implies that the log likelihood is concave which was one of the prerequisites of our algorithm (see Section 2.1). It follows that $\boldsymbol{W}(\boldsymbol{f})^{-1} = \mathrm{diag}(\exp(-\boldsymbol{f}))$.

## A.6 Pseudo-Inverse of Negative Hessian of the Log Likelihood for Multi-Class Classification

For multi-class classification (see Appendix C.3 for details), we need access to the pseudo inverse $\boldsymbol{W}^\dagger$. For this to be efficient, we derive an explicit form of $\boldsymbol{W}^\dagger$ in the following and show that matrix-vector multiplies can be implemented efficiently in $\mathcal{O}(NC)$. Since the ordering (see Appendix B.1) of $\boldsymbol{W}$ plays an important role in the derivation, we use an explicit notation in this section.

**Lemma A.7** (Explicit Pseudo-Inverse for Multi-Class Classification)
*Consider multi-class classification, such that the log likelihood $\log p(\boldsymbol{y} \mid \boldsymbol{f})$ is given by a categorical likelihood with a softmax inverse link function, then the pseudoinverse $[\boldsymbol{W}(\boldsymbol{f})]_{CN}^\dagger \in \mathbb{R}^{NC\times NC}$ of $\boldsymbol{W}(\boldsymbol{f})$ in $CN$-ordering is given by*

$$[\boldsymbol{W}(\boldsymbol{f})]_{CN}^\dagger = \begin{pmatrix} (\boldsymbol{I} - \frac{1}{C}\boldsymbol{1}\boldsymbol{1}^\top)\mathrm{diag}(\boldsymbol{\pi}_1^{-1})(\boldsymbol{I} - \frac{1}{C}\boldsymbol{1}\boldsymbol{1}^\top) & & \\ & \ddots & \\ & & (\boldsymbol{I} - \frac{1}{C}\boldsymbol{1}\boldsymbol{1}^\top)\mathrm{diag}(\boldsymbol{\pi}_N^{-1})(\boldsymbol{I} - \frac{1}{C}\boldsymbol{1}\boldsymbol{1}^\top) \end{pmatrix},$$

*where $\boldsymbol{\pi}_n = (\pi_n^1, ..., \pi_n^C)^\top \in \mathbb{R}^C$ denotes the output of the softmax for $\boldsymbol{x}_n$, i.e. $\pi_n^c := \exp(f_n^c)/\sum_{c'} \exp(f_n^{c'})$. The cost of one matrix-vector multiplication $\boldsymbol{v} \mapsto [\boldsymbol{W}(\boldsymbol{f})]_{CN}^\dagger \boldsymbol{v}$ with the pseudo-inverse is $\mathcal{O}(NC)$.*

*Proof.* By Eq. (3.38) in Rasmussen & Williams (2006), the matrix $\boldsymbol{W}(\boldsymbol{f})$ in $NC$-ordering is given by

$$[\boldsymbol{W}(\boldsymbol{f})]_{NC} = [\mathrm{diag}(\boldsymbol{\pi})]_{NC} - \boldsymbol{\Pi}\boldsymbol{\Pi}^\top,$$

where $[\text{diag}(\boldsymbol{\pi})]_{NC} = \text{diag}(\pi_1^1, \ldots, \pi_N^1, \ldots, \pi_1^C, \ldots \pi_N^C)$ and

$$\boldsymbol{\Pi} = \begin{pmatrix} \text{diag}(\pi_1^1, \ldots, \pi_N^1) \\ \vdots \\ \text{diag}(\pi_1^C, \ldots, \pi_N^C) \end{pmatrix} \in \mathbb{R}^{NC \times N}.$$

Rewriting $\boldsymbol{W}(\boldsymbol{f})$ in the $CN$-ordering, we obtain using $[\text{diag}(\boldsymbol{\pi})]_{CN} = \text{diag}(\pi_1^1, \ldots, \pi_1^C, \ldots, \pi_N^1, \ldots, \pi_N^C)$ that

$$[\boldsymbol{W}(\boldsymbol{f})]_{CN} = [\text{diag}(\boldsymbol{\pi})]_{CN} - \begin{pmatrix} \boldsymbol{\pi}_1 & & \\ & \ddots & \\ & & \boldsymbol{\pi}_N \end{pmatrix} \begin{pmatrix} \boldsymbol{\pi}_1^\top & & \\ & \ddots & \\ & & \boldsymbol{\pi}_N^\top \end{pmatrix}$$
$$= \text{blockdiag}(\text{diag}(\boldsymbol{\pi}_n) - \boldsymbol{\pi}_n \boldsymbol{\pi}_n^\top).$$

Now the pseudoinverse of a block-diagonal matrix is the block-diagonal of the block pseudoinverses, i.e. $\text{blockdiag}(\boldsymbol{A}_n)^\dagger = \text{blockdiag}(\boldsymbol{A}_n^\dagger)$ which can be shown by simply checking the definition criteria of the pseudo-inverse and using basic properties of block matrices. Therefore it suffices to show that the block pseudoinverses are given by

$$(\text{diag}(\boldsymbol{\pi}_n) - \boldsymbol{\pi}_n \boldsymbol{\pi}_n^\top)^\dagger = (\boldsymbol{I} - \frac{1}{C}\boldsymbol{1}\boldsymbol{1}^\top) \text{diag}(\boldsymbol{\pi}_n^{-1})(\boldsymbol{I} - \frac{1}{C}\boldsymbol{1}\boldsymbol{1}^\top)$$

for $n \in \{1, \ldots, N\}$. We do so by checking the definition criteria of a pseudoinverse. Let $\boldsymbol{A}_n = \text{diag}(\boldsymbol{\pi}_n) - \boldsymbol{\pi}_n \boldsymbol{\pi}_n^\top$. We begin by showing the following intermediate result:

$$\boldsymbol{A}_n(\boldsymbol{I} - \frac{1}{C}\boldsymbol{1}\boldsymbol{1}^\top) = \boldsymbol{A}_n - \frac{1}{C}(\text{diag}(\boldsymbol{\pi}_n) - \boldsymbol{\pi}_n \boldsymbol{\pi}_n^\top)\boldsymbol{1}\boldsymbol{1}^\top = \boldsymbol{A}_n - \frac{1}{C}(\boldsymbol{\pi}_n - \boldsymbol{\pi}_n(\boldsymbol{\pi}_n^\top \boldsymbol{1}))\boldsymbol{1}^\top = \boldsymbol{A}_n. \qquad (14)$$

Now let's verify the first criterion in the definition of the pseudoinverse. We have

$$\boldsymbol{A}_n(\boldsymbol{I} - \frac{1}{C}\boldsymbol{1}\boldsymbol{1}^\top) \text{diag}(\boldsymbol{\pi}_n^{-1})(\boldsymbol{I} - \frac{1}{C}\boldsymbol{1}\boldsymbol{1}^\top)\boldsymbol{A}_n = \boldsymbol{A}_n \text{diag}(\boldsymbol{\pi}_n^{-1})\boldsymbol{A}_n$$
$$= \boldsymbol{A}_n \text{diag}(\boldsymbol{\pi}_n^{-1})(\text{diag}(\boldsymbol{\pi}_n) - \boldsymbol{\pi}_n \boldsymbol{\pi}_n^\top)$$
$$= \boldsymbol{A}_n(\boldsymbol{I} - \boldsymbol{1}\boldsymbol{\pi}_n^\top)$$
$$= \boldsymbol{A}_n - (\text{diag}(\boldsymbol{\pi}_n) - \boldsymbol{\pi}_n \boldsymbol{\pi}_n^\top)\boldsymbol{1}\boldsymbol{\pi}_n^\top$$
$$= \boldsymbol{A}_n,$$

where we used (14). Next, we'll verify the second criterion.

$$(\boldsymbol{I} - \frac{1}{C}\boldsymbol{1}\boldsymbol{1}^\top) \text{diag}(\boldsymbol{\pi}_n^{-1})(\boldsymbol{I} - \frac{1}{C}\boldsymbol{1}\boldsymbol{1}^\top)\boldsymbol{A}_n(\boldsymbol{I} - \frac{1}{C}\boldsymbol{1}\boldsymbol{1}^\top) \text{diag}(\boldsymbol{\pi}_n^{-1})(\boldsymbol{I} - \frac{1}{C}\boldsymbol{1}\boldsymbol{1}^\top)$$
$$= (\boldsymbol{I} - \frac{1}{C}\boldsymbol{1}\boldsymbol{1}^\top) \text{diag}(\boldsymbol{\pi}_n^{-1})\boldsymbol{A}_n \text{diag}(\boldsymbol{\pi}_n^{-1})(\boldsymbol{I} - \frac{1}{C}\boldsymbol{1}\boldsymbol{1}^\top)$$
$$= (\boldsymbol{I} - \frac{1}{C}\boldsymbol{1}\boldsymbol{1}^\top)(\boldsymbol{\pi}_n^{-1})(\boldsymbol{I} - \frac{1}{C}\boldsymbol{1}\boldsymbol{1}^\top)$$

where we used

$$\text{diag}(\boldsymbol{\pi}_n^{-1})\boldsymbol{A}_n = \boldsymbol{I} = \boldsymbol{A}_n \text{diag}(\boldsymbol{\pi}_n^{-1}) \qquad (15)$$

as shown above. Finally, we verify the symmetry of the product of $\boldsymbol{A}_n$ and its pseudoinverse. Observe that both $\boldsymbol{A}_n$ and $(\boldsymbol{I} - \frac{1}{C}\boldsymbol{1}\boldsymbol{1}^\top) \text{diag}(\boldsymbol{\pi}_n^{-1})(\boldsymbol{I} - \frac{1}{C}\boldsymbol{1}\boldsymbol{1}^\top)$ are symmetric. Therefore we have

$$(\boldsymbol{A}_n(\boldsymbol{I} - \frac{1}{C}\boldsymbol{1}\boldsymbol{1}^\top) \text{diag}(\boldsymbol{\pi}_n^{-1})(\boldsymbol{I} - \frac{1}{C}\boldsymbol{1}\boldsymbol{1}^\top))^* = (\boldsymbol{I} - \frac{1}{C}\boldsymbol{1}\boldsymbol{1}^\top) \text{diag}(\boldsymbol{\pi}_n^{-1})(\boldsymbol{I} - \frac{1}{C}\boldsymbol{1}\boldsymbol{1}^\top)\boldsymbol{A}_n$$

and

$$((\boldsymbol{I} - \frac{1}{C}\boldsymbol{1}\boldsymbol{1}^\top) \text{diag}(\boldsymbol{\pi}_n^{-1})(\boldsymbol{I} - \frac{1}{C}\boldsymbol{1}\boldsymbol{1}^\top)\boldsymbol{A}_n)^* = \boldsymbol{A}_n(\boldsymbol{I} - \frac{1}{C}\boldsymbol{1}\boldsymbol{1}^\top) \text{diag}(\boldsymbol{\pi}_n^{-1})(\boldsymbol{I} - \frac{1}{C}\boldsymbol{1}\boldsymbol{1}^\top).$$

Thus if we can show that $\boldsymbol{A}_n$ and $(\boldsymbol{I} - \frac{1}{C}\mathbf{1}\mathbf{1}^\top) \operatorname{diag}(\boldsymbol{\pi}_n^{-1})(\boldsymbol{I} - \frac{1}{C}\mathbf{1}\mathbf{1}^\top)$ commute we have shown the remaining symmetry criteria of the pseudoinverse. It holds that

$$\boldsymbol{A}_n(\boldsymbol{I} - \frac{1}{C}\mathbf{1}\mathbf{1}^\top) \operatorname{diag}(\boldsymbol{\pi}_n^{-1})(\boldsymbol{I} - \frac{1}{C}\mathbf{1}\mathbf{1}^\top) \stackrel{(14)}{=} \boldsymbol{A}_n \operatorname{diag}(\boldsymbol{\pi}_n^{-1})(\boldsymbol{I} - \frac{1}{C}\mathbf{1}\mathbf{1}^\top) \stackrel{(15)}{=} (\boldsymbol{I} - \frac{1}{C}\mathbf{1}\mathbf{1}^\top)$$

as well as

$$(\boldsymbol{I} - \frac{1}{C}\mathbf{1}\mathbf{1}^\top) \operatorname{diag}(\boldsymbol{\pi}_n^{-1})(\boldsymbol{I} - \frac{1}{C}\mathbf{1}\mathbf{1}^\top)\boldsymbol{A}_n \stackrel{(14)}{=} (\boldsymbol{I} - \frac{1}{C}\mathbf{1}\mathbf{1}^\top) \operatorname{diag}(\boldsymbol{\pi}_n^{-1})\boldsymbol{A}_n \stackrel{(15)}{=} (\boldsymbol{I} - \frac{1}{C}\mathbf{1}\mathbf{1}^\top)$$

This completes the proof for the form of the pseudoinverse. For the complexity of multiplication, note that multiplying with $(\boldsymbol{I} - \frac{1}{C}\mathbf{1}\mathbf{1}^\top) \operatorname{diag}(\boldsymbol{\pi}_n^{-1})(\boldsymbol{I} - \frac{1}{C}\mathbf{1}\mathbf{1}^\top)$ has cost $\mathcal{O}(C)$, since it decomposes into two multiplications with $(\boldsymbol{I} - \frac{1}{C}\mathbf{1}\mathbf{1}^\top)$ which is linear and one elementwise scaling. Therefore the cost of multiplication with the pseudoinverse consisting of $N$ blocks has complexity $\mathcal{O}(NC)$. □

## B   Implementation Details

### B.1   Ordering within Vectors & Matrices

**Ordering within Vectors.** By default, we assume all vectors and matrices to be represented in $CN$-ordering. For example, the mean vector was introduced as the aggregated outputs of the mean function $m \colon \mathbb{X} \to \mathbb{R}^C$ for all data points $\boldsymbol{m} = m(\boldsymbol{X}) = (m(\boldsymbol{x}_1)^\top, \dots, m(\boldsymbol{x}_N)^\top)^\top$. With $m(\boldsymbol{x}_n)^\top = (m_n^1, \dots, m_n^C)$ denoting the $C$ outputs for data point $\boldsymbol{x}_n$, we can write $\boldsymbol{m}$ as $\boldsymbol{m} = (m_1^1, m_1^2, \dots, m_1^C, \dots, m_N^1, m_N^2, \dots, m_N^C)$. We call that representation $CN$-ordering because the superscript $c$ moves *first* and the subscript $n$ moves *second*. Consecutively, $(m_1^1, m_2^1, \dots, m_N^1, \dots, m_1^C, m_2^C, \dots, m_N^C)$ corresponds to $NC$-ordering.

**Ordering within Matrices.** The same terminology can be applied to matrices, where the rows and columns can be represented in $CN$ or $NC$-ordering. Depending on the context, different representations are beneficial. For example, in $CN$-ordering, $\boldsymbol{W}$ is block-diagonal (due to our iid assumption, see Section 2.1) with $N$ blocks of size $C \times C$ on the diagonal. In contrast, when the $C$ outputs of the hidden function are assumed to be independent of each other, $\boldsymbol{K}$ is block diagonal only in $NC$-ordering. So, based on the chosen ordering, different structures arise that we can exploit in subsequent computations (e.g. when we compute the inverse of $\boldsymbol{W}$, see Appendix B.3).

### B.2   Stopping Criteria in Algorithms 1 and 2

**Stopping Criterion in Algorithm 1.** The OUTERSTOPPINGCRITERION() we use for our experiments is based on the *relative change* of the vector $\boldsymbol{g}_i = \boldsymbol{f}_i - \boldsymbol{m}$. When $\|\boldsymbol{g}_i - \boldsymbol{g}_{i-1}\| \|\boldsymbol{g}_i\|^{-1} \leqslant \delta$ falls below the convergence tolerance $\delta$ (by default, $\delta = 1\%$), the loop over $i$ terminates. Of course, other convergence criteria are also conceivable. Depending on the application one might want to customize the criterion and, for example, include the marginal uncertainty at the training data.

**Stopping Criterion in Algorithm 2.** We use the same INNERSTOPPINGCRITERION() as in (Wenger et al., 2022b, Section S3.2): The loop over $j$ terminates if the norm of the residual $\|\boldsymbol{r}_j\| < \max\{\delta_{\text{abs}}, \delta_{\text{rel}}\|\hat{\boldsymbol{y}} - \boldsymbol{m}\|\}$ is below an absolute tolerance $\delta_{\text{abs}}$ or below the scaled norm of the right-hand side $\hat{\boldsymbol{y}} - \boldsymbol{m}$ of the linear system. By default, both tolerances are set to $10^{-5}$. Additionally, we typically specify a maximum number of iterations. The solver is also terminated when the normalization constant $\eta_j \leqslant 0$. This can happen due to numerical imprecision if the linear system is badly conditioned, e.g. if some eigenvalues of the linear system are close to zero.

### B.3   Cost Analysis of IterNCGP

In this section, we investigate the computational costs of ITERNCGP in more detail. We start with a discussion of the computational costs for matrix-vector products with $\boldsymbol{K}$, $\boldsymbol{W}^{-1}$ and $\boldsymbol{C}_j$ and then analyze the runtime and memory costs of the individual algorithms (Algorithms 1 to 3).

### B.3.1 Matrix-Vector Products

ITERGP is an *iterative matrix-free* algorithm and our algorithm ITERNCGP inherits that property: The matrices $\boldsymbol{K}$, $\boldsymbol{W}^{-1}$ and $\boldsymbol{C}_j$ are evaluated lazily, i.e. matrix-vector products are evaluated *without* forming the matrices in memory explicitly. This enables our algorithm to scale to problems where a naive approach causes memory overflows. In Algorithms 1 to 3, the memory and runtime cost for matrix-vector products with $\boldsymbol{K}$ are denoted by $\mu_{\boldsymbol{K}}$ and $\tau_{\boldsymbol{K}}$ and by $\mu_{\boldsymbol{W}^{-1}}$ and $\tau_{\boldsymbol{W}^{-1}}$ for products with $\boldsymbol{W}^{-1}$.

**Products with $\boldsymbol{K}$.** Matrix-vector products with $\boldsymbol{K}$ can be decomposed into products with its sub-matrices. The associated memory costs $\mathcal{O}(\mu_{\boldsymbol{K}})$ can thereby be reduced basically arbitrarily and the runtime can be improved by using specialized software libraries such as KEOPS (Charlier et al., 2021) and parallel hardware (i.e. GPUs). Still, products with $\boldsymbol{K}$ are computationally relatively expensive, since this operation is typically *quadratic* in the number of training data points $N$.

**Products with $\boldsymbol{W}^{-1}$ (General Case).** Under the assumptions on the likelihood from Section 2.1, $\boldsymbol{W}$ is block-diagonal with $N$ blocks of size $C \times C$ (in $CN$-ordering, see Appendix B.1). Here, we denote these blocks by $\boldsymbol{W}_1, \ldots, \boldsymbol{W}_N$. It can be easily verified that $\boldsymbol{W}^{-1}$ is also a block-diagonal matrix and the blocks on its diagonal are the inverses of $\boldsymbol{W}_1, \ldots, \boldsymbol{W}_N$.

Consider the matrix-vector product $\boldsymbol{v} \mapsto \boldsymbol{W}^{-1}\boldsymbol{v} =: \boldsymbol{w} \in \mathbb{R}^{NC}$. In the vectors $\boldsymbol{v}$ and $\boldsymbol{w}$, we repeatedly group $C$ consecutive entries which results in segments $\boldsymbol{w}_n, \boldsymbol{v}_n \in \mathbb{R}^C$ for $n = 1, ..., N$, i.e.

$$\underbrace{\begin{pmatrix} \boldsymbol{W}_1^{-1} & & \\ & \ddots & \\ & & \boldsymbol{W}_N^{-1} \end{pmatrix}}_{=\boldsymbol{W}^{-1}} \cdot \underbrace{\begin{pmatrix} \boldsymbol{v}_1 \\ \vdots \\ \boldsymbol{v}_N \end{pmatrix}}_{=\boldsymbol{v}} = \underbrace{\begin{pmatrix} \boldsymbol{w}_1 \\ \vdots \\ \boldsymbol{w}_N \end{pmatrix}}_{=\boldsymbol{w}}.$$

It holds that $\boldsymbol{w}_n = \boldsymbol{W}_n^{-1}\boldsymbol{v}_n$, i.e. each segment in $\boldsymbol{w}$ is the product of a single $C \times C$ block from $\boldsymbol{W}^{-1}$ with one segment from $\boldsymbol{v}$. Computing $\boldsymbol{w}_n$ thus amounts to solving a linear system of size $C$ with cost $\mathcal{O}(C^3)$. The total cost for all $N$ segments is thus $\mathcal{O}(NC^3)$. However, the $N$ linear systems are independent of each other and can thus be solved in parallel. So, if appropriate computational resources are available, the total runtime complexity can be reduced to $\mathcal{O}(C^3)$.

In general, $\boldsymbol{W}^{-1}$ requires $\mathcal{O}(NC^2)$ in terms of memory consumption. If needed, these costs can be reduced further to $\mathcal{O}(C^2)$ because (as explained above), products with $\boldsymbol{W}^{-1}$ can be broken down into products with the individual blocks of $\boldsymbol{W}^{-1}$. We can perform those products sequentially such that only a single block is present in memory at a time.

**Products with $\boldsymbol{W}^{-1}$ (Special Cases).** In many cases, we can multiply with $\boldsymbol{W}^{-1}$ more efficiently. In the multi-class classification case, the runtime and memory costs for multiplication with the pseudo inverse $\boldsymbol{W}^\dagger$ can be reduced to $\mathcal{O}(NC)$ (see Appendix A.6). In the regression case ($C = 1$), $\boldsymbol{W}^{-1}$ is a diagonal matrix of size $N \times N$. The memory and runtime costs are thus in $\mathcal{O}(N)$. An example is Poisson regression, for which we derive the explicit form of $\boldsymbol{W}^{-1}$ in Appendix A.5.

**Products with $\boldsymbol{C}_j$.** $\boldsymbol{C}_j = \boldsymbol{Q}_j \boldsymbol{Q}_j^\top$ is represented via its matrix root $\boldsymbol{Q}_j \in \mathbb{R}^{NC \times B}$. This allows for efficient storage and matrix-vector multiplies $\boldsymbol{v} \mapsto \boldsymbol{C}_j \boldsymbol{v} = \boldsymbol{Q}_j(\boldsymbol{Q}_j^\top \boldsymbol{v})$ in $\mathcal{O}(BNC)$.

### B.3.2 Cost Analysis Algorithms 1 to 3

The runtime and memory complexity for the operations in Algorithms 1 to 3 is given directly in the pseudo code. Here, we provide some additional information for the costs that depend on the user's choices and put the costs of the individual algorithms into perspective.

**Algorithm 2 (IterGP).** The runtime cost for selecting an action $\mathcal{O}(\tau_{\text{POLICY}})$ depends on the underlying policy. For Cholesky actions ($\boldsymbol{s}_j = \boldsymbol{e}_j$) or CG ($\boldsymbol{s}_j = \boldsymbol{r}_{j-1}$), the runtime cost is insignificant since no additional computations are required at all.

One iteration's total computational cost (without prediction) is dominated by two matrix-vector products with $\boldsymbol{K}$ in terms of runtime and $\mathcal{O}(BNC)$ in terms of storage requirements (for the buffers $\boldsymbol{S}$ and $\boldsymbol{T}$ as well

as the matrix root $\boldsymbol{Q}_j$). The *initial* size (i.e. the number of columns) of $\boldsymbol{S}$, $\boldsymbol{T}$ and $\boldsymbol{Q}_j$ is given by the rank limit $R$ used in Algorithm 3. Henceforth, one column is added to each of the buffers and matrix root in *each* solver iteration, increasing their size to $B = R + j$ in iteration $j$. It is thus reasonable to include an upper bound on the iteration number in the stopping criterion of Algorithm 2.

**Algorithm 3 (Virtual Solver Run with Optional Compression).** The total runtime complexity of Algorithm 3 is $\mathcal{O}(B\tau_{\boldsymbol{W}^{-1}} + B^2 NC)$, i.e. dominated by matrix-matrix products involving the buffers and $\boldsymbol{W}^{-1}$. In terms of memory requirements, the buffers $\boldsymbol{S}$, $\boldsymbol{T}$, and $\boldsymbol{Q}_0$ are the decisive contributors with $\mathcal{O}(RNC)$. The truncation of the eigendecomposition provides a way to control that bound by resetting the current buffer size $B$ to an arbitrary number $R \leqslant B$. In comparison to Algorithm 2, the computational costs are practically of minor importance since no multiplications with $\boldsymbol{K}$ are necessary.

**Algorithm 1 (IterNCGP Outer Loop).** The costs $\mathcal{O}(\tau_{\boldsymbol{m}})$ for evaluating $m$ on the training data depends on the choice of mean function. For a constant mean function, no computations are necessary, so runtime costs are negligible. This can be different e.g. for applications in Bayesian deep learning, where evaluating $m$ requires forward passes through a neural network.

## C    Experimental Details

Throughout the paper, we use binary classification as an illustrative and supporting example (Figures 1 to 4). The two main experiments follow in Section 5: Poisson regression (Section 5.1, Figure 5) and large-scale GP multi-class classification (Section 5.2, Figure 6). In the following, we provide additional details for all those experiments.

### C.1    Binary Classification

**Binary Classification with *one* latent function.** Consider a binary classification task, i.e. $C = 2$. Being able to report the probability for *one* of the two classes is sufficient because they have to add up to one for every data point. Thus, while $C = 2$, $N$-dimensional vectors are typically used to describe this problem (Rasmussen & Williams, 2006, Section 3.4). Using only a single latent function is convenient for illustrative purposes, as e.g. the action vectors $\boldsymbol{s}$ in Algorithm 2 are $N$-dimensional (not $2N$-dimensional) and thus easier to visualize.

**1D Data.** We use a one-dimensional training set in Figure 2. $\boldsymbol{X}$ is created by sampling $N = 50$ data points between $-3$ and $5$. The hidden function $f$ is a draw from a GP with mean zero and a GPyTorch (Gardner et al., 2018) RBF kernel with `lengthscale = 1.0` and `outputscale = 5.0`. For each datapoint $\boldsymbol{x}_n$, we sample the positive label with probability logistic($f(\boldsymbol{x}_n)$).

**2D Data.** Two-dimensional data is used in Figures 1, 3 and 4. The data-generating process is analogous to the 1D data, only now, the $N = 100$ training inputs are in the 2D plane: The first coordinate is sampled uniformly between $-3$ and $5$, the second between $-4$ and $1$. The hyperparameters of the RBF kernel are `lengthscale = 1.0`, `outputscale = 10.0` for Figures 3 and 4 and `outputscale = 20.0` for Figure 1.

**Details Figure 1.** In this figure, we compare two versions of our algorithm: IterNCGP-Chol without recycling and IterNCGP-CG with recycling and with compression ($R = 10$). Both runs were conducted on a CPU. The computation of the NLL loss is *not* included in the runtime measurement. A description of how the NLL loss can be computed for arbitrary $C$ is given in Appendix C.3.

**Details Figure 2.** For Figure 2, we compute a sequence of *precise* Newton steps by using IterNCGP with unit vector actions and $j \leqslant N$ solver iterations. Note that the Newton linear system is $N$-dimensional, i.e. we actually obtain $\boldsymbol{f}_i$ as defined by Equation (3).

**Details Figure 3.** In this plot, we compare unit vector actions (IterNCGP-Chol) and residual actions (IterNCGP-CG) for the first Newton step ($i = 0$) at three solver iterations $j \in \{1, 10, 19\}$. The true posterior mean function $m_{0,*}$ and covariance function $K_{0,*}$ are computed by using IterNCGP-Chol and $j \leqslant N = 100$ iterations. Figure 8 shows the covariance functions corresponding to the mean functions in Figure 3.

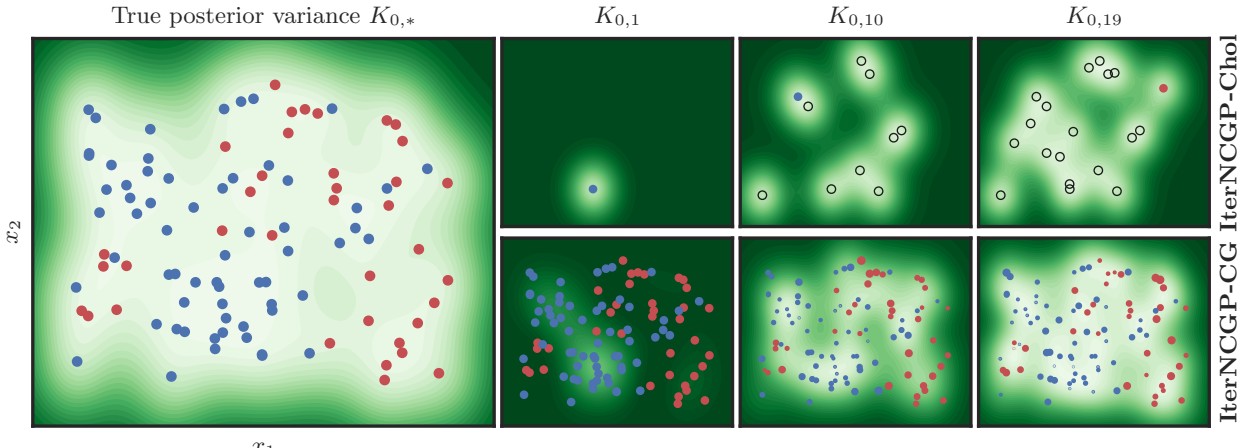

**Figure 8: Different Policies of IterNCGP Applied to GP Classification.** *(Left)* The true posterior covariance $K_{0,*}$ (  ) for a binary classification task (●/●). *(Right)* Current estimate of the posterior covariance after $1, 10$, and $19$ iterations with the unit vector policy (*Top*) and the CG policy (*Bottom*). Shown are the data points selected by the policy in this iteration with the dot size indicating their relative weight. For IterNCGP-Chol, data points are targeted one by one and previously used data points are marked with (○).

The actions are visualized by scaling the training data points according to their *relative weight*: First, we take the absolute value of the action vector $\boldsymbol{s}$ from Algorithm 2 (element-wise) and then scale its entries linearly such that the smallest entry is 0 and the largest is 1.

**Details Figure 4.** In this figure, we show the initial mean function $m_{1,0}$ and covariance function $K_{1,0}$ in the *second* Newton step for different buffer sizes $R \in \{0, 1, 3, 10\}$. We use CG actions for the first Newton step and let the solver run until convergence (this takes 19 iterations).

## C.2 Poisson Regression

In Section 5.1, we apply IterNCGP to Poisson regression to demonstrate our algorithm's ability to generalize to other (log-concave) likelihoods and to explore the trade-off between the number of (outer loop) mode-finding steps and (inner loop) solver iterations.

**Poisson Likelihood.** We consider count data $\boldsymbol{y} \in \mathbb{N}_0^N$ that is assumed to be generated from a Poisson likelihood with unknown positive rate $\lambda \colon \mathbb{X} \to \mathbb{R}_+$. Modeling $\lambda$ with a GP which may take positive *and negative* values, would therefore be inappropriate. However, we can use a GP for the log rate $f \colon \mathbb{X} \to \mathbb{R}$ and regard this as the unknown latent function. With $\boldsymbol{\lambda} \coloneqq \lambda(\boldsymbol{X}) = \exp(f(\boldsymbol{X})) = \exp(\boldsymbol{f})$, the likelihood is given by

$$p(\boldsymbol{y} \mid \boldsymbol{f}) = \prod_{n=1}^N \frac{\lambda_n^{y_n} \exp(-\lambda_n)}{y_n!}.$$

The gradient and (inverse) Hessian of the log likelihood can be derived in closed form, see Appendix A.5.

**Data & Model.** First, we create $\boldsymbol{X}$ by linearly spacing $N = 100$ points between 0 and 1. For the count data $\boldsymbol{y}$, we sample from a GP with zero mean and a GPyTorch (Gardner et al., 2018) RBF-kernel with `lengthscale = 0.1` and `outputscale = 5.0`. That GP $f$ represents the log-Poisson rate. We then draw counts from a Poisson distribution with rate $\lambda(\boldsymbol{x}_n) = \exp(f(\boldsymbol{x}_n))$ for each data point in the training set. In this experiment, we conduct multiple IterNCGP runs on different training sets. These sets are created by re-drawing from the Poisson distributions with the same rates, i.e. the underlying GP for the log rate does *not* change. Our NCGP's prior uses the same RBF kernel to avoid model mismatch.

**IterNCGP-CG Approaches.** We consider IterNCGP-CG with four different schedules: A fixed budget of 100 iterations is distributed uniformly over $5, 10, 20$ or $100$ outer loop steps (see Algorithm 1), which limits the number of inner loop iterations (see Algorithm 2) to $j \leqslant 20, 10, 5$ or $1$. For each schedule, we perform 10 runs with different training sets, see above. Each run uses recycling without compression. For this experiment, the convergence tolerance in Algorithm 1 is set to $\delta = 0.001$. All runs are performed on a single NVIDIA GeForce RTX 2080 Ti 12 GB GPU.

**Tracking of Performance Metrics.** As a performance metric, we use the NLL loss. The computation of the NLL loss for the test and training set is *not* included in the runtime reported in the results. For the NLL loss, we approximate the integral from Section 2.3 with MC samples: For each test datum $\boldsymbol{x}_\diamond$, we draw $10^5$ MC samples from $\mathcal{N}(m_{i,j}(\boldsymbol{x}_\diamond), K_{i,j}(\boldsymbol{x}_\diamond, \boldsymbol{x}_\diamond))$, map those samples $\{f_{\diamond,k}\}_{k=1}^{10^5}$ through the likelihood $p(y_\diamond \mid f_{\diamond,k})$ and average. This yields a loss value for $\boldsymbol{x}_\diamond$ and we obtain the training/test NLL loss by averaging these loss values for all data points in the training/test set.

**Approximate Rate Distribution.** Using IterNCGP-CG for the Poisson regression problem results in a sequence of posteriors $\mathcal{GP}(m_{i,j}, k_{i,j})$. By drawing MC samples from those posterior GPs and mapping them through the exponential, we obtain an approximated (skewed) distribution for the rate $\lambda$. In Figure 5 *(Right)*, we report its median and a 95 % confidence interval between the 2.5 % and 97.5 % percentile.

## C.3 Large-Scale GP Multi-Class Classification

In this experiment, we empirically evaluate IterNCGP on a large-scale GP multi-class classification problem to exhibit its scalability. We also investigate the impact of compression on performance.

**Data.** We consider a Gaussian mixture problem with $C = 10$ classes in 3D. For each class, we sample a mean vector uniformly in $[-1, 1]^3$ and a positive definite covariance matrix. For the covariance matrix, we first create a $3 \times 3$ matrix $\boldsymbol{C}$ with entries between 0 and 1 (sampled uniformly) and compute the eigenvectors $\boldsymbol{U}$ of $\boldsymbol{C}\boldsymbol{C}^\top$. Then, we create three eigenvalues $\{\lambda_d\}_{d \in \{1,2,3\}}$ uniformly between 0.001 and 0.1 and form the covariance matrix from the eigenvectors $\boldsymbol{U}$ and these eigenvalues, i.e. $\boldsymbol{U}\operatorname{diag}(\lambda_1, \lambda_2, \lambda_3)\boldsymbol{U}^\top$. For each class, $10^4$ data points are sampled from the respective Gaussian distribution. This amounts to $N = 10^5$ data points in total. For testing, $N_\diamond = 10^4$ data points are used ($10^3$ per class).

Our benchmark (Figure 6) uses multiple runs for each method. The runs differ in the seed that is used to sample from the Gaussians, i.e. the training and test set are different for each run (the underlying Gaussians distributions remain the same). Both the training and test set used in the first run are shown in Figure 9.

**Model.** We use a softmax likelihood (see Appendix A.6 for the details on the pseudo inverse $\boldsymbol{W}^\dagger$) and assume independent GPs for the $C$ outputs of the latent function. Each of these GPs uses the zero function as the prior mean and a Matérn($\frac{3}{2}$) kernel. We use the KeOps (Charlier et al., 2021) version of the GPyTorch (Gardner et al., 2018) kernel with `lengthscale = 0.05` and `outputscale = 0.05`.

**SoD Approaches.** For the SoD approaches, we create a random subset of the training data (sampling without replacement) of a specific subset size $N_{\text{sub}}$. We then explicitly form $\hat{\boldsymbol{K}}(\boldsymbol{f}_i)$ for every Newton step and compute its Cholesky decomposition via PyTorch's (Paszke et al., 2019) `torch.linalg.cholesky` (instead of using IterGP in Algorithm 1 to ensure a competitive baseline implementation). In our experiment, we use four different subset sizes $N_{\text{sub}} \in \{250, 500, 1000, 2000\}$. Each setting is run five times with different random seeds (see above).

**SVGP.** SVGP (Hensman et al., 2015; Titsias, 2009) is a commonly used variational method for approximative inference in non-conjugate GPs. We use GPyTorch's (Gardner et al., 2018) SVGP implementation and optimize the ELBO for $10^4$ seconds using Adam with batch size $= 1024$. The learning rate $\alpha \in \{0.001, 0.01, 0.05\}$ and the number of inducing points $U \in \{1000, 2500, 5000, 10000\}$ are tuned via grid search (using only a single run). We use $U/C$ inducing points per class ($C = 10$ classes) and initialize them as a random subset of the training data. Within the given runtime budget, SVGP performs between 6000 ($U = 1000$) and 600 ($U = 10000$) epochs.

For each of the 12 runs, we extract 6 performance indicators: lowest training/test NLL loss during training, highest training/test accuracy during training and training/test expected calibration error (ECE) at the very

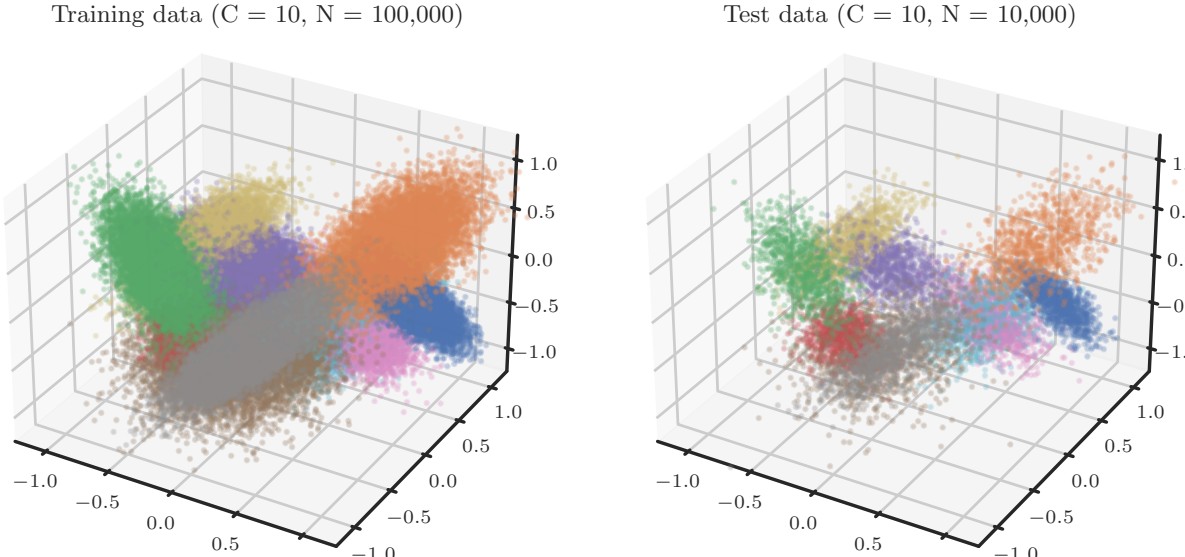

Training data (C = 10, N = 100,000)          Test data (C = 10, N = 10,000)

**Figure 9: Gaussian Mixture Training and Test Data.** Training data *(Left)* and test data *(Right)* for the first run. Note that the underlying Gaussians are the same for training and test set and for all runs.

end of training. For each of those 6 categories, we determine those two runs with the best performance. This procedure results in a set of 3 runs in total, that are considered the "best" runs for SVGP. Only for those 3 settings, we perform 5 runs each and report their mean performance in Figure 6.

**IterNCGP-CG Approaches.** For comparison, we apply our matrix-free algorithm IterNCGP with residual actions to the *full* training set. We use two configurations: The first one uses recycling *without* compression (i.e. $R = \infty$), the second one uses recycling *with* compression ($R = 10$). The number of solver iterations per step is limited by $j \leq 5$. For both settings, we perform 5 runs with different random seeds (see above) and report their mean performance in Figure 6.

**Tracking of Performance Metrics.** As performance metrics, we use accuracy, the negative log likelihood (NLL) loss and the expected calibration error (ECE) (Kumar et al., 2019) on both the training and test set. The computation of those six metrics is *not* included in the runtime reported in the results (Figure 6). First, we compute the predictive mean $m_{i,j}(\boldsymbol{x}_\diamond)$ and marginal variance $\mathrm{diag}(K_{i,j}(\boldsymbol{x}_\diamond, \boldsymbol{x}_\diamond))$ (see Equations (8) and (9)) for each test input $\boldsymbol{x}_\diamond$. Then, we use the probit approximation (MacKay, 1992; Spiegelhalter & Lauritzen, 1990) to obtain the predictive probabilities

$$\boldsymbol{\pi}_\diamond = \mathrm{softmax}\left(\frac{m_{i,j}(\boldsymbol{x}_\diamond)}{\sqrt{1 + \frac{\pi}{8}\mathrm{diag}(K_{i,j}(\boldsymbol{x}_\diamond, \boldsymbol{x}_\diamond))}}\right) \in \mathbb{R}^C,$$

where the vector division is defined element-wise. This is an approximation of the integral from Section 2.3. All three performance metrics are based on the predictive probabilities.

- **Accuracy.** The prediction for $\boldsymbol{x}_\diamond$ is given by $\arg\max_c([\boldsymbol{\pi}_\diamond]_c)$, i.e. by the class with the largest predictive probability. The accuracy is defined as the ratio of correctly classified data.

- **NLL Loss.** The NLL loss for $\boldsymbol{x}_\diamond$ is defined as the log-probability for the actual class $y_\diamond$, i.e. $\log([\boldsymbol{\pi}_\diamond]_{y_\diamond})$. We obtain the NLL training and test loss by averaging the individual loss values for the entire training/test set.

- **ECE.** The expected calibration error (ECE) (Kumar et al., 2019) is a measure for the calibration of the predictive probabilities. It groups the probabilities of the predicted classes (i.e. the classification confidences) into bins and, within these bins, compares the average confidence with the actual

accuracy. We use `MulticlassCalibrationError` from `torchmetrics` (Detlefsen et al., 2022) with default parameter `n_bins=15`.

**Individual runs.** Figure 6 shows the *average* performance for each of the nine methods/variants over five runs. In order to show, that the observed performance differences are not due to chance, we show the *individual* runs in Figure 10.

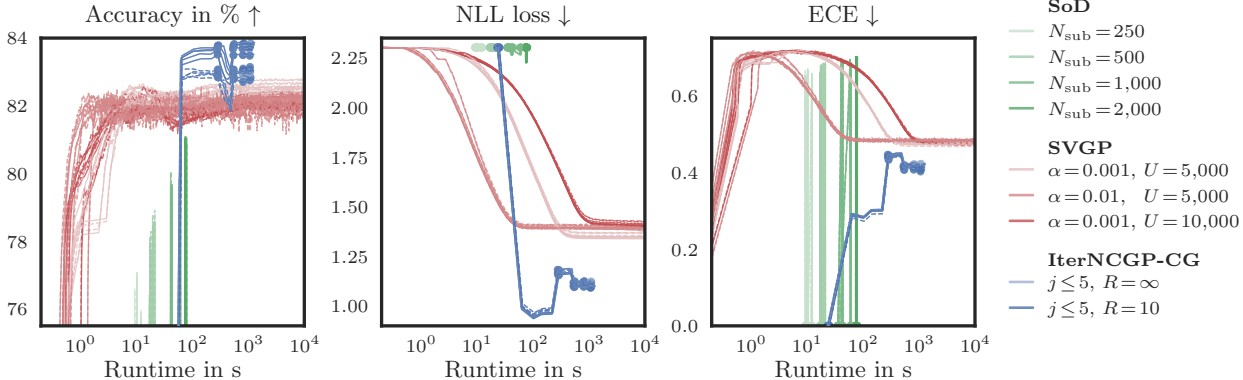

**Figure 10: Large-Scale GP Classification.** Same as Figure 6, but showing all runs individually (instead of just their average).

## C.4 GP Multi-Class Classification on MNIST

To demonstrate IterNCGP's applicability to real-world datasets, we perform an additional experiment (similar to the experiment described in Appendix C.3) on a subset ($N_{\text{sub}} = 20{,}000$) of the MNIST (Lecun et al., 1998) dataset. In the following, we describe the experiment and results in more detail.

**Remark.** For the experiment on synthetic data, we use KeOps on top of GPyTorch for fast kernel-matrix multiplies. However, KeOps scales poorly with the data dimension ($D = 28^2 = 784$ for MNIST)[5]. We therefore revert to GPyTorch's standard implementation. This implementation of the kernel matrix-vector product is fast but causes out-of-memory errors for large datasets, which is why we limit our benchmark to $N_{\text{sub}} = 20{,}000$ training data. To fully realize the potential of our method using KeOps, it might be advisable to apply it only to problems with data dimensions smaller than around 100.

**Data & Model.** We use 20,000 training and 10,000 test images from the MNIST dataset and the softmax likelihood. Our model for the latent function is a multi-output GP which uses $C = 10$ independent GPs, each of which is equipped with a Matérn($\frac{3}{2}$) kernel.

**Kernel Hyperparameters.** As a first step, we determine suitable hyperparameters (the outputscale and lengthscale) for the Matérn($\frac{3}{2}$) kernel by running GPyTorch's SVGP implementation. We use 1000 inducing points per class, a batch size of 1024 and optimize the ELBO using ADAM with learning rate 0.001 for 30 epochs (this results in `lengthscale = 1.550934` and `outputscale = 0.451591`). Note that choosing hyperparameters with SVGP may give SVGP an advantage in what performance it can reach, making it a competitive baseline.

**SVGP and IterNCGP-CG Approaches.** We compare IterNCGP-CG and SVGP both using the same fixed kernel hyperparameters (see above). IterNCGP-CG is applied with recycling and $R = \infty$, i.e. without compression. We exclude compression since both IterNCGP runs converge within three iterations, see below. The number of inner loop iterations is limited by $j \leqslant 1$ or $j \leqslant 5$, i.e. two runs are performed. For the SVGP approach, we optimize the ELBO using ADAM with batch size 1024 for 200 seconds. The number of inducing points and the learning rate are tuned via grid search. As in Appendix C.3, we use

---

[5]https://www.kernel-operations.io/keops/_auto_benchmarks/plot_benchmark_high_dimension.html (accessed May 2024)

$U \in \{1000, 2500, 5000, 10000\}$ inducing points and three different learning rates $\alpha \in \{0.001, 0.01, 0.05\}$. All 14 runs are performed on a single NVIDIA GeForce RTX 2080 Ti 12 GB GPU.

**Results.** The results are shown in Figure 7. They show the two IterNCGP runs and the best four SVGP runs (these include the best two runs for each of the six performance metrics training/test accuracy/NLL/ECE). Our observations are mostly consistent with the results on the synthetic data (Figure 6): Both IterNCGP runs significantly outperform the best SVGP runs in terms of NLL loss and accuracy. Only the ECE achieved by IterNCGP is slightly worse than for SVGP. However, as explained in Section 5.2, a small ECE on its own is not conclusive since we can easily construct a classifier with perfect ECE by randomly sampling predictions.

