# OpenReview forum: "Accelerating Non-Conjugate Gaussian Processes By Trading Off Computation For Uncertainty"
_TMLR — Accepted by TMLR_

### Review · Reviewer_oipv · 2024-12-03

**Summary Of Contributions:**

This paper proposes IterNCGP, a method for computation-aware non-conjugate Gaussian process regression via the Laplace approximation. The method first interprets the Laplace approximation for GPs as sequential GP solves, then applies the recently proposed IterGP for computation-aware solving. Between the Newton steps of the Laplace approximation, a "virtual solver" is proposed to precondition linear solves using previously explored actions, providing significant performance boosts. To further reduce computational and (mostly) memory costs, the use of a truncated representation of previous actions is justified and used. The proposed method is then applied to Poisson regression and multi-class classification, where it shows extremely time-efficient performance.

**Audience:**

Yes

**Claims And Evidence:**

Yes

**Requested Changes:**

## Critical
- My only critical change requested is in the abstract, where it is claimed that "our method significantly accelerates training compared to competitive baselines [..]". I understand that "training" is an overloaded term in GPR, but am afraid this claim could be misinterpreted as including hyperparameter optimization. I believe this is demonstrated only when "training" means "posterior (predictive) inference," and the claim should be disambiguated accordingly.

## Minor
- Please consider changing Figure 1, or otherwise explaining the structure of the algorithm in the introduction, and including Algorithms 2 and 3 in the main text.
- Please consider adding some discussion regarding hyperparameter selection.
- Please consider including further baselines, such as the SVGP with NGD.
- Please discuss some of the limitations regarding the Laplace approximation.

I also have some smaller comments:
- Equation (4) uses $f_{i+1}$, but I think this should be $f_i$.
- Footnote 4 uses "Eqn. (10)", which is written as "Eq. (10)" in other places.
- The outer and inner loops use (arbitrarily different) different`StoppingCriterion()` functions, but this is not reflected in pseudocode. Semantically, it seems more correct to specify, e.g., `OuterStoppingCriterion()` and `InnerStoppingCriterion()`.
- On page 6, the sentence "This virtual solver run [...] orders of magnitude cheaper, than running the solver [...]" shouldn't have a comma.
- In Appendix B.1, "$CN$-orderingbecause" is missing a space.
- In Appendix C.3, for the probit approximation, Spiegelhalter & Lauritzen (1990) is also a standard reference.

# References

Dutordoir, V., Durrande, N., & Hensman, J. (2020, November). Sparse Gaussian processes with spherical harmonic features. In International Conference on Machine Learning (pp. 2793-2802). PMLR.

Khan, M., & Lin, W. (2017, April). Conjugate-computation variational inference: Converting variational inference in non-conjugate models to inferences in conjugate models. In Artificial Intelligence and Statistics (pp. 878-887). PMLR.

Kuss, M., Rasmussen, C. E., & Herbrich, R. (2005). Assessing Approximate Inference for Binary Gaussian Process Classification. Journal of Machine Learning Research, 6(10).

Nickisch, H., & Rasmussen, C. E. (2008). Approximations for binary Gaussian process classification. Journal of Machine Learning Research, 9(10), 2035-2078.

Salimbeni, H., Eleftheriadis, S., & Hensman, J. (2018, March). Natural gradients in practice: Non-conjugate variational inference in Gaussian process models. In International Conference on Artificial Intelligence and Statistics (pp. 689-697). PMLR.

Spiegelhalter, D. J., & Lauritzen, S. L. (1990). Sequential updating of conditional probabilities on directed graphical structures. _Networks_, _20_(5), 579-605.

**Strengths And Weaknesses:**

## Strengths

The contribution to Gaussian processes is significant, providing extremely efficient computation in GPC/non-conjugate GPR inference. The approach is elegant and theoretically justified, and the mathematical derivations and proofs in the appendices are clear and easy to follow. The paper is generally well-written and does a good job of explaining some complex ideas, even if some non-standard background is necessary.

## Weaknesses
Overall, this is a very nice paper; I think addressing the weaknesses below would strengthen the paper, but did not find any critical issues.

**No Proposed Hyperparameter Selection**
The issue of hyperparameter selection is treated as out-of-scope, but this seems like a rather big hurdle for the application of the proposed method. This is especially relevant as simultaneously learning the hyperparameters and variational distribution is an advantage of SVGPs when considering "performance vs. time" in a more realistic scenario that includes hyperparameter optimization.

While I think this work has plenty of theoretical merits without including hyperparameter selection, this is a notable practical issue and merits some discussion. Even if no satisfying approach is immediately available, highlighting this as future work might be worthwhile.

**Presentation of Algorithms**
The main algorithms, the inner and outer loop of IterNCGP, are only presented in the appendix. This caused some page-turning for me in an otherwise smooth read, as the algorithms are often referenced in the main text. I think it would improve the presentation to include Algorithms 2 and 3 in the main text.

**Figure 1 Placement**
I found Figure 1 more confusing than I found it motivating before reading the rest of the paper. In particular, the inner loop-outer loop structure is not yet discussed, which makes $i$ and $j$ somewhat difficult to understand. It is also not yet explained that the method approximates the Laplace approximation, which makes "Newton steps" opaque. I would encourage the authors to find ways to make this "marquee" figure a bit more approachable, either by changing the figure or by explaining the algorithm structure at a high level in the introduction.

Even something relatively brief like "the algorithm consists of an 'outer loop,' which iteratively solves approximate Newton steps of the Laplace approximation via an 'inner loop,' which uses probabilistic numerics to account for the uncertainty in computationally-limited solving" would help in understanding what "the panels show the marginal uncertainty over the latent function at Newton step $i$ and solver iteration $j$" means.

**Baseline Methods**
While I find the experiments convincing, I think they could be made stronger by comparing IterNCGP to some more modern methods. For example, incorporating natural gradients in various ways may result in significantly better time vs. performance curves (Khan & Lin, 2017; Salimbeni et al. 2018). Using natural gradient descent, in particular, is already implemented in gPyTorch and should be relatively easy to test against. Besides natural gradients, some other works report substantial improvements to the wall-clock time of GP training (e.g., Dutordoir et al. (2020)), though I understand these are a bit more specialized. It would be interesting to consider the performance gains relative to these methods.

**Limitations of the Laplace Approximation**
The accuracy of the Laplace approximation fundamentally limits the accuracy of the proposed method. While this is nicely acknowledged and briefly discussed in the conclusion, it would perhaps be beneficial to discuss this a bit more, especially since some older results suggest the Laplace approximation to be quite poor in the context of small-scale GPC (with Kuss et al. (2005) going so far as to say "the Laplace approximation is so inaccurate that we advise against its use, especially when predictive probabilities are to be taken seriously"; also see Nickisch & Rassmussen (2008)). I think the experimental results of this paper show this is not necessarily a concern when in the modern "big data" regime, but this could be pointed out as counter to the traditional logic.

---

> ### Author Response · Authors · 2024-12-13
>
> Dear Reviewer oipv,
>
> Thank you so much for taking the time to review our manuscript and for providing such
> positive and constructive feedback. We greatly appreciate your concrete and actionable
> suggestions to further strengthen the paper!
>
> Once all reviews are available, we will address your concerns and carefully incorporate
> your suggestions, alongside any others we may receive, into a revised submission.
>
> Thanks again for your time and effort!

---

> ### Author Response · Authors · 2025-02-14
>
> Dear Reviewer oipv,
>
> Thanks again for your great feedback and the many thoughtful suggestions to further
> strengthen our paper. We have carefully considered all your comments and incorporated
> them, together with the feedback from the other reviewers, into a revised manuscript. We
> have summarized all the changes we made in a joint response to all reviewers, which we
> have just posted.
>
> Thanks again for your time and effort!

---

> ### Comment · Reviewer_oipv · 2025-02-18
>
> Thank you to the authors for effectively addressing most of the mentioned weaknesses. I believe the only one not addressed was including stronger baseline methods, which I still think would strengthen the paper, but is not necessary in my opinion to meet the TMLR criteria for publication.
>
> My only other (critical) concern is that the newly added reference to Kass (1990) does not seem correct to me. If the version I have access to is the same, there are two missing authors (Tierney and Kadane), and the book title should be capitalized.

---

> > ### Author Response · Authors · 2025-02-19
> >
> > Dear Reviewer oipv,
> >
> > > My only other (critical) concern is that the newly added reference to Kass (1990) does not seem correct to me. If the version I have access to is the same, there are two missing authors (Tierney and Kadane), and the book title should be capitalized.
> >
> > Thanks for pointing this out! We have just uploaded a revised version of the PDF in which these errors have been corrected.

---

### Review · Reviewer_Fjue · 2025-01-16

**Summary Of Contributions:**

In this paper, the authors propose an extension to the IterGP method (Wenger et al., 2023) for likelihoods that are not Gaussian. Their method is based on the Laplace approximation, obtaining a Gaussian approximation of the posterior centered on a mode of the true posterior, and uses Newton's method to find an approximation to a mode. As an extension of IterGP, the authors manage to view this procedure under the lens of probabilistic numerics, allowing the numerical approximations to contribute to the posterior variance of their approximation, therefore increasing the quality of the approximated uncertainty quantification.

The first contribution of the authors is to show the identification between performing the Newton steps of MAP estimation and the posterior computation of the approximate GP posterior for each step. Under this identification, the IterGP framework can be applied at each Newton step, thus adding a contribution to the posterior uncertainty that comes from approximating the matrix inversions required in the steps. The authors formalize this and show their derivations in the appendix.

Their second contribution comes from the notion of "recycling" and "compression," where some computation required in each Newton step can be reused between iterations and, moreover, can be compressed by truncating the SVD decomposition of the recycled computation.

The authors test the correctness and speed of their algorithm in two synthetic datasets and show promising results.

**Audience:**

Yes

**Broader Impact Concerns:**

I believe this work does not require a broader impact statement, as I could not think of ethical issues that could be raised with it's deployment and application.

**Claims And Evidence:**

Yes

**Requested Changes:**

- Possible typo in Section 2.3: Equations (4) and (5) might have problems with indices. $\boldsymbol{f}\_{i+1}$ should probably be $\boldsymbol{f}\_i$ in Eq. (4) since it's stated that $\boldsymbol{f}\_{i}\approx\boldsymbol{f}\_{\mathrm{MAP}}$.
- The paragraph "Newton's Method as Sequential GP Regression" is hard to understand: As better explained in Appendix A.1, it is the case that the posterior under iteration $i$ and the computation of the next iteration of the Newton step are obtained by computing the representer weights of the fictitious regression problem with targets $\hat{\boldsymbol{y}}\_i$. Unless this information is presented, it is not immediately clear what is meant by equating Newton's Method to computing the posterior sequential GP regression, after all, for NM, one obtains a vector $\boldsymbol{f}\_{i+1}$ while the other gives a distribution.
- Figure 6 is missing a line for $R=\infty$. I couldn't find the metrics for the light-colored blue line, and the text does not clarify why it is missing from the figure.
- [Critical] More explicitly acknowledge that the current methodology does not provide a clear recipe for kernel hyperparameter estimation in the limitations.
- [Critical] Incorporate MNIST experiments in the main paper. As the authors still have two more pages of content available, I believe there is no reason for the results of this experiment to be mentioned in the main paper but not included in it.

**Strengths And Weaknesses:**

Strengths:

- The method is well inspired and shows the benefit of incorporating additional sources of uncertainty into GP modeling.
- Connecting Newton steps with sequential GP regression is interesting and can even open new avenues of exploration for different methods using Laplace approximation.
- The arguments of the authors are well-supported and connected to literature in numerics and probabilistic methods, thus increasing the overall familiarity of the community with these disciplines.

Weaknesses:

- The paper is overall well-written; however, I believe key parts could be more clearly explained to better convey what exactly the authors are proposing. I also believe that a short review of IterGP may be warranted in the background section as the proposed method is so heavily based on it.
- It is well-known that the perfomance of GP methods heavily depend on the hyperparameters of their kernels and likelihoods. Most of the mainstream methods do hyperparameter selection using fast gradient-based optimization, instead of costly grid/random search with cross-validation. In the MNIST experiment, the hyperparameters estimated using the ELBO of variational methods is copied over to be used in IterNCGP and no clear way to estimate these hyperparameters using only IterNCGP is given. This limitation should be better communicated to the reader as it might hinder the applicability of the method in practice.
- The empirical evaluation of the method does not go significantly beyond synthetic data. A short additional experiment on a subsampled 30% of MNIST is only shown in details in the appendix. As an example, the paper "Large-Scale Cox Process Inference using Variational Fourier Features" by John and Hensman (2018) includes datasets with an order of magnitude more data in lower dimensions while using well-known variational methods.

---

> ### Author Response · Authors · 2025-01-24
>
> Dear Reviewer Fjue,
>
> Thank you for the time and effort you dedicated to reviewing our paper and for your thoughtful suggestions! We will address your concerns and revise our paper according to your requested changes once all reviews are available.
>
> Thanks again for your valuable input!

---

> ### Author Response · Authors · 2025-02-14
>
> Dear Reviewer Fjue,
>
> Thank you once again for your thoughtful and constructive review. We have carefully
> incorporated all reviewers' feedback into the revised version of our paper. The changes
> we made are summarized in a joint response to all reviews that we have just posted. We
> sincerely hope that our revisions meet your expectations and address all of your
> concerns satisfactorily. If you have any further questions or comments, please do not
> hesitate to contact us.
>
> We truly appreciate your time and effort—thank you again!

---

### Review · Reviewer_uNBX · 2025-02-04

**Summary Of Contributions:**

The authors introduce a method for accelerating Laplace approximations of GP models by extending IterGP for non-Gaussian likelihoods. The speedup comes from using matrix-free computation techniques in the optimization literature and posing Newton's method as a form of GP regression.

**Audience:**

Yes

**Claims And Evidence:**

Yes

**Requested Changes:**

Refer to the weaknesses in the previous section. In summary, I think the paper is quite nice but the authors should spend some time rewriting Sec. 3 to be a more gentler transition from introducing the problems of non-conjugate GP regression to introducing their method.

**Strengths And Weaknesses:**

Strengths
---
1. I think the paper is a neat idea, and I quite like the connection of GPs with optimization.
2. Generally speaking, the paper is well written.
3. The results show that your method is considerably faster compared to typical sparse variational approaches, which is nice to see.

Weaknesses
---
1. To understand the paper, one needs to be first quite familiar with Wenger's paper. It would be better to write this paper without totally assuming familiarity, otherwise the paper reads as a further appendix to Wenger et al. instead of a stand-alone paper. This is related to the next point as well.
2. Section 3 is not so easy to understand, particularly in terms of the language used regarding "actions" and "policy". I understand this is the terminology used in the Wenger paper but it doesn't seem that helpful (in my opinion) to allude to reinforcement learning if none of your applications are in RL. I would introduce the concepts of action and policy, along with the examples in Sec. 3.2 first before introducing Eq. 10.
3. If you look at the GP from the weight space view, it seems straightforward to obtain a Laplace approximation in the non-Gaussian case. Why did you not do this? Especially since you mention the Trippe et al. paper is a a weight space-variant of your paper,
4. It seems like you use $i$ and $j$ to be iteration and policy indices as well as the total number of iterations and policy size in Sec. 3? (ex: $S_j = (s_1, \ldots, s_j)$ has rank $j$). But then it seems you use $B$ to denote the policy size later? Please be consistent with the notation.
5. Why do you need to further compress the $S$ and $T$ further to have rank $R$ if the computation in Alg. 1 is dominated by the eigendecomposition of a user-selected size $B$ matrix? If the size $B$ is already an issue, shouldn't you just set $B$ to be whatever $R$ would be?
6. The method seems more involved than how it is presented in Alg.1, and many of the important details are pushed to the Algorithms in the appendix. I think they should be moved to the main text and you should spend some time guiding the reader through the implementation. Moreover, the grey subtitles of Alg. 3 are not super clear to me as to what they really mean so it would be helpful to explain some of the matrices and vectors defined here a little bit more.
7. I would be interested to see how this method fares on real data classification problems with a considerably large number of classes (CIFAR-100, for example) in comparison to the other GP methods, since the class size seems to cause problems here.

---

> ### Author Response · Authors · 2025-02-14
>
> Dear Reviewer uNBX,
>
> Thanks a lot for your valuable feedback, your suggestions to improve our work, and for
> the thoughtful questions you raised. We have written a joint comment in which we address
> the main changes we have made to the paper in response to the reviews. Here, we want to
> specifically address your questions.
>
> > If you look at the GP from the weight space view, it seems straightforward to obtain a
> > Laplace approximation in the non-Gaussian case. Why did you not do this? Especially
> > since you mention the Trippe et al. paper is a a weight space-variant of your paper,
>
> An equivalent weight-space characterization of our approach only exists if we assume a
> finite number of features (as in Trippe et al.), meaning a parametric kernel of the form
> $k(x_0, x_1) = \phi(x_0)^\top \Sigma \phi(x_1)$, where $\phi$ is a feature map. As we
> write in the paper, we believe the approach in Trippe et al. to be an instantiation of
> our framework, where the choice of actions determines the choice of low-rank
> approximation (e.g., a randomized SVD). However, the function-space/kernel perspective
> allows one to handle arbitrarily many features (even infinitely many, e.g., for Matérn
> kernels). This means that our derivation and approach are more generally applicable, as
> they do not require the kernel of the latent Gaussian process to be a parametric kernel.
> In cases where one can evaluate the kernel independently of the feature dimension ($D$
> in Trippe et al.'s notation), our approach is also more efficient since it has
> complexity $\mathcal{O}(N^2M)$ rather than $\mathcal{O}(NDM)$ (again using Trippe et
> al.'s notation from Table 1 and their assumption that $D \geq N$).
>
> > Why do you need to further compress the $S$ and $T$ further to have rank $R$ if the
> > computation in Alg. 1 is dominated by the eigendecomposition of a user-selected size
> > $B$ matrix? If the size $B$ is already an issue, shouldn't you just set $B$ to be
> > whatever $R$ would be?
>
> $B$ denotes the size of the buffers $S \in \mathbb{R}^{NC \times B}$ and $T \in
> \mathbb{R}^{NC \times B}$. With every inner-loop iteration of the probabilistic linear
> solver, an additional column is appended to $S$ and $T$. That means, after $I$ Newton
> steps with $J$ inner-loop iterations each, the buffers $S$ and $T$ steadily grow to size
> $B = I \cdot J$ if *no* compression is used. For large-scale problems, this can be
> prohibitive in terms of memory consumption. By using compression, we can control the
> size of the buffers by keeping only the most relevant $R$ actions at the beginning of
> the Newton step. The size of the buffer is thus limited by $B \leq R + J$, i.e.,
> importantly, it is *not* a function of the number of Newton iterations $I$.
>
> Thanks again for your review! We hope that you found our answers satisfactory. If you
> have any follow-up questions, please do not hesitate to contact us.

---

> > ### Comment · Reviewer_uNBX · 2025-03-01
> >
> > Hi, thank you for your reply.
> >
> > I still have one more question about the weight-space characterization: It still seems like an apt comparison if you compare a finite basis function expansion of the GP with your method though. Essentially, you're still relying on some low-rank approximations for your method, so finite basis function should be equivalent to a low-rank approximation to the kernel in function-space? Regarding the choice of kernel, you could rely on some kernel approximations with basis function expansions (ex: Random Fourier Features (Rahimi and Recht) or BANK (Oliva et al.)) that should approximate whatever kernel you choose (based on whatever prior you place on the random features and should have complexity of $O(N^2M)$). In this setting you could obtain a Laplace approximation directly, and perform something similar to your actions during optimization (subsampling, weighting by residuals, etc.).

---

> > > ### Author Response · Authors · 2025-03-06
> > >
> > > Dear Reviewer uNBX,
> > >
> > > Thank you for your interesting comment!
> > >
> > > As you correctly point out, directly approximating the kernel via a finite basis function expansion (e.g. RFF) leads to a parametric kernel of the form $k(x_0, x_1) = \phi(x_0)^\top \Sigma \phi(x_1)$, or equivalently a low-rank approximation to the kernel matrix. One could then apply the approach of Trippe et al. (2019), which as we write above, we then believe to be an instantiation of our framework for a specific choice of actions.
> > >
> > > Our framework is not equivalent to using a finite set of basis functions to approximate the kernel, since we **only approximate the downdate** in the posterior covariance (see Eq. (9)) with a low-rank matrix. In particular, the posterior mean is a linear combination of $N$ kernel functions $K(\cdot, X)$, rather than a linear combination of $D$ features and the prior covariance function appears as is in the posterior covariance function (Eq. (9)), rather than being approximated with a finite basis expansion. This is also why we do not observe variance starvation away from the data, as has been observed for RFF (see Figure 1 of Wang et al. 2018), but rather a reversion to the prior variance. This can be seen directly from the downdate form of the posterior covariance function in (Eq. (9)), where the downdate term vanishes for a test point sufficiently far from the data, assuming a stationary kernel.
> > >
> > > Finally, note that in our experiments we are already comparing to a low-rank approximation of the kernel in the form of SVGP, which uses a Nyström approximation of the kernel (see Section 3.2 of Wild et al. (2021)) given by
> > > $$
> > >     k(x, x') = k(x, Z)k(Z, Z)^{-1}k(Z, x')
> > > $$
> > > for a set of $D< N$ inducing points $Z$, which in our experiments are learned from the data.
> > >
> > > References:
> > > - Zi Wang, Clement Gehring, Pushmeet Kohli, and Stefanie Jegelka. Batched large-scale Bayesian optimization in high-dimensional spaces. *International Conference on Artificial Intelligence and Statistics (AISTATS)* 2018. URL https://arxiv.org/abs/1706.01445.
> > > - Veit Wild, Motonobu Kanagawa, and Dino Sejdinovic. Connections and equivalences between the Nyström method and sparse variational Gaussian processes. arXiv pre-print, 2021. URL http://arxiv.org/ abs/2106.01121.

---

### Author Response · Authors · 2025-02-14

Dear Reviewers,

Thank you so much for your constructive feedback and thoughtful questions! In response
to your suggestions, we have made the following updates to our paper (the references we
provide below refer to this updated version):
- **Abstract:** We replaced "training" with "posterior inference" [oipv].
- **Section 1:** In the last paragraph of the introduction, we briefly explain the
  two-loop structure of our algorithm before referencing Figure 1 [oipv].
- **Section 2.3:** We have clarified why $f_{i+1}$ is correct in Equation (4) by being
  more explicit about the local Laplace approximation [oipv, Fjue].
- **Section 3.1:** We have revised the entire section to improve the clarity of the
  derivation of our method.
    - We added an "Overview" paragraph at the beginning of Section 3.1 to provide a
      high-level summary of the key points, better guiding the reader through the
      content [uNBx, Fjue].
    - We clarified the connection between the Newton step and the posterior predictive
      mean by elaborating on the common underlying linear system of equations [Fjue].
    - We give a general introduction to IterGP and already introduce and explain the
      terms "action" and "policy" here [uNBx, Fjue].
    - Both Algorithms 1 (outer loop) and 2 (inner loop) have been moved from the
      appendix to the main text [uNBx, oipv].
- **Section 5.2:** The MNIST experiment is now part of the main text (see Figure 7)
  [Fjue].
- **Section 6:** The conclusion has been expanded to include a more thorough discussion
  of the limitations of our work. In particular, we acknowledge that the current
  methodology does not provide a ready-to-use solution for hyperparameter tuning and
  discuss how this could be addressed in future work [oipv, Fjue]. We have also expanded
  our discussion of the Laplace approximation [oipv].

We have also made the following minor improvements:
- **Algorithms 1 and 2:** We improved the clarity of the comments and provide more
  details about the vectors and matrices in the text (e.g., we now explain the role of
  the representer weights in more detail in Section 3.1) [uNBx].
- **Figure 6:** The two IterNCGP-CG variants are almost identical, making the light blue
  curve barely visible. We added a remark in the caption to clarify this [Fjue].
- We have also addressed all items under "smaller comments" in oipv's review.

These updates, we believe, have further strengthened the paper, addressing key points
raised in the reviews. We appreciate the constructive evaluations and feel encouraged by
the positive feedback!

---

### Decision · Action_Editor_kt9D · 2025-04-15

**Recommendation:** Accept as is

**Comment:**

I and the reviewers agree that the paper presents a well-thought-out idea with clear practical benefits in scalability. The reviewers feel the authors have done a good job of improving the clarity of the paper. While the reviewers noted that they would like to see more datasets and baselines in the experiment, the benefit of the method is sufficiently demonstrated and I would not make this a condition of acceptance.

**Audience:**

Yes - Gaussian processes are of wide interest, and scalability remains a problem.

**Claims And Evidence:**

Yes, the claims are supported by experimental evidence.